# RCB initiates *Arabidopsis* thermomorphogenesis by stabilizing the thermoregulator PIF4 in the daytime

Yongjian Qiu [1,2,3✉], Elise K. Pasoreck [1,3], Chan Yul Yoo [1,3], Jiangman He [1], He Wang[1], Abhishesh Bajracharya [2], Meina Li[1], Haley D. Larsen [1], Stacey Cheung[1] & Meng Chen [1✉]

Daytime warm temperature elicits thermomorphogenesis in *Arabidopsis* by stabilizing the central thermoregulator PHYTOCHROME INTERACTING transcription FACTOR 4 (PIF4), whose degradation is otherwise promoted by the photoreceptor and thermosensor phytochrome B. PIF4 stabilization in the light requires a transcriptional activator, HEMERA (HMR), and is abrogated when HMR's transactivation activity is impaired in *hmr-22*. Here, we report the identification of a *hmr-22* suppressor mutant, *rcb-101*, which surprisingly carries an A275V mutation in REGULATOR OF CHLOROPLAST BIOGENESIS (RCB). *rcb-101/hmr-22* restores thermoresponsive PIF4 accumulation and reverts the defects of *hmr-22* in chloroplast biogenesis and photomorphogenesis. Strikingly, similar to *hmr*, the null *rcb-10* mutant impedes PIF4 accumulation and thereby loses the warm-temperature response. *rcb-101* rescues *hmr-22* in an allele-specific manner. Consistently, RCB interacts directly with HMR. Together, these results unveil RCB as a novel temperature signaling component that functions collaboratively with HMR to initiate thermomorphogenesis by selectively stabilizing PIF4 in the daytime.

[1] Department of Botany and Plant Sciences, Institute for Integrative Genome Biology, University of California, Riverside, CA, USA. [2] Department of Biology, University of Mississippi, Oxford, MS, USA. [3]These authors contributed equally: Yongjian Qiu, Elise K Pasoreck, Chan Yul Yoo. ✉email: yqiu@olemiss.edu; meng.chen@ucr.edu

The sensation of temperature changes is essential for the survival of plants. In angiosperms (flowering plants), such as the reference species *Arabidopsis thaliana* (*Arabidopsis*), a shift in ambient growth temperature of only a few degrees can significantly alter the expression of hundreds of temperature-responsive genes, resulting in dramatic adaptive responses in plant development, growth, metabolism, and immunity; these responses are collectively referred to as thermomorphogenesis[1,2]. Because increases in global temperature are expected to drastically reduce crop productivity[3,4], understanding the mechanism of temperature signaling has become imminent to create a knowledge base for devising strategies to sustain crop production in a changing climate[5].

Plants sense changes in ambient temperature via phytochrome B (PHYB)[6,7]. PHYB belongs to a small family of red (R) and far-red (FR) photoreceptors in *Arabidopsis* that includes five members, PHYA-E[8]. PHYs monitor changes in light quality, quantity, and periodicity through photoconversions between two relatively stable conformers, an R-light-absorbing inactive Pr and an FR-light-absorbing active Pfr conformer[9,10]. In addition to photoconversion, the active Pfr can spontaneously revert to the inactive Pr in a light-independent process called dark or thermal reversion[11]. The rate of thermal reversion of PHYB, in particular, is rapid enough to rival that of photoactivation and accelerates with temperature increases of between 10 and 30 °C[6]. These intrinsic properties of the PHYB molecule enable the activity of PHYB to respond to changes in ambient temperature[6,7], making PHYB a thermosensor in addition to a photoreceptor. Because warm temperatures often coincide with high light intensities during the daytime—a combined light and temperature condition where a significant amount of steady-state PHYB remains in the active form[12]—the essence of understanding thermomorphogenesis is to elucidate how warm temperatures engage with PHYB signaling.

A well-established experimental paradigm to interrogate PHYB-mediated light signaling is de-etiolation—a developmental transition that occurs when young seedlings emerge from the ground and first encounter light[13]. *Arabidopsis* seedlings germinated and grown in darkness (mimicking the condition of underground growth) adopt a dark-grown developmental program called skotomorphogenesis or etiolation, which promotes the elongation of the embryonic stem (hypocotyl) and inhibits leaf development and chloroplast biogenesis. Exposing dark-grown seedlings to light initiates de-etiolation to reprogram seedlings to a light-dependent developmental program called photomorphogenesis, which attenuates hypocotyl elongation, stimulates leaf expansion, and promotes chloroplast biogenesis[8,14]. De-etiolation displays visibly trackable readouts such as hypocotyl elongation and leaf greening, reporting two major downstream functions of PHYB signaling: the control of plant growth and chloroplast biogenesis, respectively[8,14]. The effectiveness of PHYB in restraining hypocotyl elongation has also been widely used as a physiological assay for PHYB-mediated thermomorphogenesis because warm temperatures significantly accelerate hypocotyl elongation[6,7,15,16]. Making it more complicated, hypocotyl elongation is gated by the circadian clock and partitioned to different times in short-day (SD) and long-day (LD) conditions[17–19]. In SD conditions, hypocotyl elongation occurs mainly at the end of the night or in the dark, when PHYB has mostly reverted to the inactive Pr[18]. By contrast, in LD conditions including continuous light, hypocotyl elongation peaks during the daytime, when PHYB is in the active Pfr[17–19]. Consistently, warm temperatures enhance hypocotyl growth accordingly at different times between SD and LD conditions[20,21]. This discrepancy in the timing of hypocotyl growth between nighttime and daytime represents two mechanistic strategies by which temperature cues engage with PHYB signaling in different growing seasons. Therefore, although hypocotyl elongation under both SD and LD conditions can be strongly influenced by temperature changes, the underpinning mechanisms are distinct[20,22].

PHYB controls seedling morphogenesis primarily by regulating a family of basic helix-loop-helix transcription factors called PHYTOCHROME-INTERACTING FACTORs (PIFs), which includes eight members: PIF1, PIF3-8, and PIL1 (PIF3-Like1)[23,24]. PIFs accumulate to high levels in dark-grown seedlings and act collectively to repress photomorphogenesis by promoting hypocotyl elongation and blocking leaf development and chloroplast biogenesis[25–28]. Different PIFs perform overlapping and distinct roles[25,26]. For example, PIF1, PIF3, PIF4, PIF5, and PIF7 promote hypocotyl growth by activating growth-relevant genes, such as those involved in the biosynthesis and signaling of the plant growth hormone auxin[25,26,29–31]. PIF1, PIF3, and PIF5 inhibit chloroplast biogenesis by repressing photosynthesis-associated nuclear- and plastid-encoded genes[26,27,32–36]. PHYB controls the activities of PIFs at multiple levels. During de-etiolation, photoactivated PHYB in the nucleus induces photomorphogenesis primarily by promoting ubiquitin–proteasome-dependent degradation of PIF1, PIF3, PIF4, and PIF5[35,37–40]. However, interestingly, PIF4 and PIF5 (but not PIF1 and PIF3), together with PIF7—whose protein level is not significantly reduced during de-etiolation[41]—can accumulate in the daytime when seedlings grow in diurnal or continuous light conditions[22,42–45]. In particular, PIF4 and PIF7 are required for thermomorphogenesis, and their protein levels are further elevated by warm temperatures[22,45–47]. In SD conditions, the SD-specific induction of *PIF4* expression at the end of the night coincides with the disappearance of active PHYB due to thermal reversion, allowing PIF4 to accumulate to high levels to promote hypocotyl elongation before dawn[18,20,48]. In striking contrast, in LD conditions, *PIF4* transcripts accumulate only during the daytime[18,20,48]. Because, even in elevated temperatures, a significant amount of PHYB during the daytime stays in the Pfr form that mediates PIF4 degradation[12], a mechanism must be implemented to stabilize PIF4 or modulate PHYB-mediated PIF4 degradation.

How PIF4 is stabilized during the daytime remains poorly understood. We have previously shown that daytime PIF4 stabilization by warm temperatures depends on a transcriptional activator, HEMERA (HMR)[22]. HMR is a nuclear and plastidic dual-targeted protein required for PHYB-mediated photomorphogenesis and thermomorphogenesis[22,49–51]. While plastidic HMR (also called pTAC12) is an essential component of the plastid-encoded RNA polymerase responsible for the expression of plastid-encoded photosynthesis genes[52], nuclear HMR is a transcriptional activator that directly interacts with PHYB and all PIFs[51]. Intriguingly, HMR exerts opposing effects on the stability of the two closely related transcriptional regulators, PIF3 and PIF4. HMR facilitates PIF3 degradation in photomorphogenesis[49,51,53] but promotes PIF4 stabilization in thermomorphogenesis[22]. Both roles of HMR rely on its 9-amino-acid transcription activation domain (9aaTAD)[22,51]. A weak loss-of-function *hmr* allele, *hmr-22*, which carries a D516N mutation in HMR's 9aaTAD, blocks PIF3 degradation, PIF4 accumulation, and the activation of a subset of light- and temperature-responsive PIF target genes, suggesting an intimate relationship between the transcriptional activity of HMR and the stability of PIF3 and PIF4[22,51]. To further understand the mechanism of PIF4 stabilization by warmer temperatures in the daytime, we performed a forward genetic screen for *hmr-22* suppressors that can revert *hmr-22*'s defects in thermomorphogenesis. Here, we report the first *hmr-22* suppressor mutant and the identification of a new signaling partner of HMR necessary for stabilizing PIF4 in thermomorphogenesis.

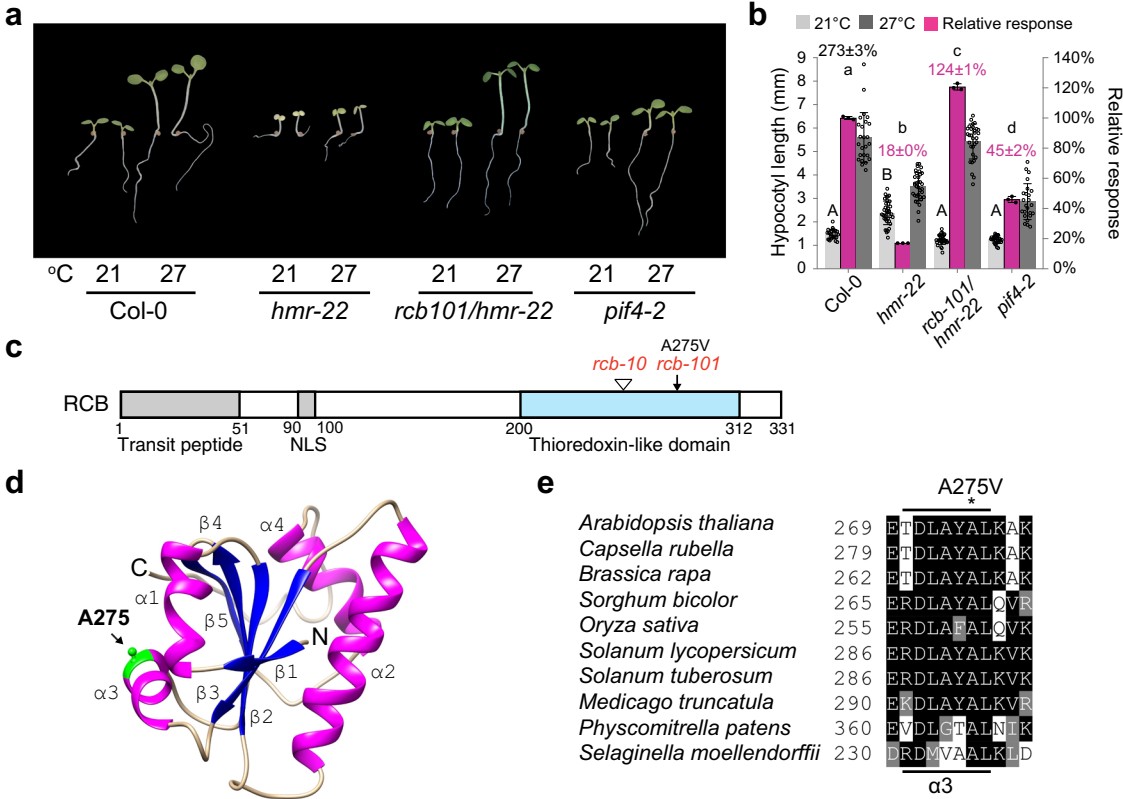

**Fig. 1 Identification of *rcb-101* as a suppressor of *hmr-22* in thermomorphogenesis. a** Representative images of 4-d-old Col-0, *hmr-22*, *rcb-101/hmr-22*, and *pif4-2* seedlings grown in 50 μmol m$^{-2}$ s$^{-1}$ R light at either 21 or 27 °C. **b** Hypocotyl length measurements of the seedlings in (**a**). The light- and dark-gray bars represent hypocotyl length measurements at 21 °C and 27 °C, respectively. The percent increase in hypocotyl length (mean ± s.d., $n = 3$ biological replicates) of Col-0 at 27 °C is shown in black above its columns. The magenta bars show the relative response, which is defined as the relative hypocotyl response to 27 °C of a mutant compared with that of Col-0 (set at 100%). Error bars for the hypocotyl measurements represent the s.d. ($n > 30$ seedlings); error bars for the relative responses represent the s.d. of three biological replicates. The centers of the error bars represent the mean values. Purple numbers show the mean ± s.d. values of relative responses and different lowercase letters denote statistically significant differences in relative responses (ANOVA, Tukey's HSD, $p < 0.01$, $n = 3$ biological replicates). Different uppercase letters denote statistically significant differences in hypocotyl length at 21 °C (ANOVA, Tukey's HSD, $p < 0.01$, $n > 24$ seedlings). **c** Schematic of the domain structure of RCB and the mutations in *rcb-101* and *rcb-10*. NLS, nuclear localization signal. **d** Simulated structure of RCB's thioredoxin-like domain highlighting Ala-275 in α3[36]. **e** Amino acid sequence alignment of selected RCB orthologs showing that Ala-275 is highly conserved in land plants. The underlying source data of the hypocotyl measurements in (**b**) are provided in the Source Data file.

## Results

**Identification of *rcb-101* as a *hmr-22* suppressor.** To investigate the mechanism of HMR-mediated PIF4 stabilization under warm daytime temperatures, we mutagenized *hmr-22* using ethyl methanesulfonate (EMS) and carried out a forward genetic screen for second-site suppressor mutants that could restore *hmr-22*'s defect in hypocotyl elongation in continuous R light at 27 °C. We chose to perform our screen in 50 μmol m$^{-2}$ s$^{-1}$ R light, a relatively high light intensity, because we observed an even greater difference in hypocotyl length of Col-0 between 21 and 27 °C under the stronger light condition compared with the previously used 10 μmol m$^{-2}$ s$^{-1}$ R light (Fig. 1a, b)[22]. One suppressor mutant, which we named *rcb-101* (explained below), completely rescued the short-hypocotyl phenotype of *hmr-22* at 27 °C and even displayed an enhanced thermal response of 124% compared with the wild-type Col-0 (Fig. 1a, b). In contrast to the pale-green phenotype of *hmr-22*, *rcb-101/hmr-22* was noticeably greener, indicative of a rescue of *hmr-22*'s defect in chloroplast biogenesis as well (Fig. 1a).

To map the *rcb-101* mutation, we crossed *rcb-101/hmr-22* (in Col-0) to a null *hmr-1* allele in L*er* background to generate an F2 mapping population, wherein only the *hmr-22* allele (but not the wild-type *HMR*) was present, thereby allowing the identification

of the *rcb-101* suppressor mutant in either homozygous or hemizygous *hmr-22* background. Using 507 F2 lines with the suppressor phenotypes, we mapped the *rcb-101* mutation to a 70-kb region on chromosome 4 (Supplementary Fig. 1). Interestingly, the region contains the *RCB* (*Regulator of Chloroplast Biogenesis*, At4g28590) gene, which encodes a recently reported light signaling component required for both PHYB signaling in the nucleus as well as PHYB-mediated nucleus-to-plastid signaling for the control of chloroplast biogenesis[27,36]. After sequencing the *RCB* locus in *rcb-101/hmr-22*, we found a C-to-T mutation at nucleotide 14125627 in *RCB*'s third exon, resulting in an A275V substitution in RCB (Fig. 1c). The C-to-T mutation co-segregated with the suppressor phenotype in all 1014 recombinant chromosomes in the mapping population (Supplementary Fig. 1), strongly supporting that it is the causal mutation for the *rcb-101/hmr-22* suppressor phenotypes. Thus, this suppressor mutant was named *rcb-101*.

RCB contains a C-terminal thioredoxin (Trx) fold that lacks the canonical C-X-X-C catalytic motif for Trx's reductase activity but maintains a prototypical βαβαβαββα secondary structural arrangement similar to that of *E. coli* Trx[27,36]. Alanine 275 resides in α3 that connects the N- and C-terminal halves of the Trx fold (Fig. 1d), and it is highly conserved in RCB orthologs across land

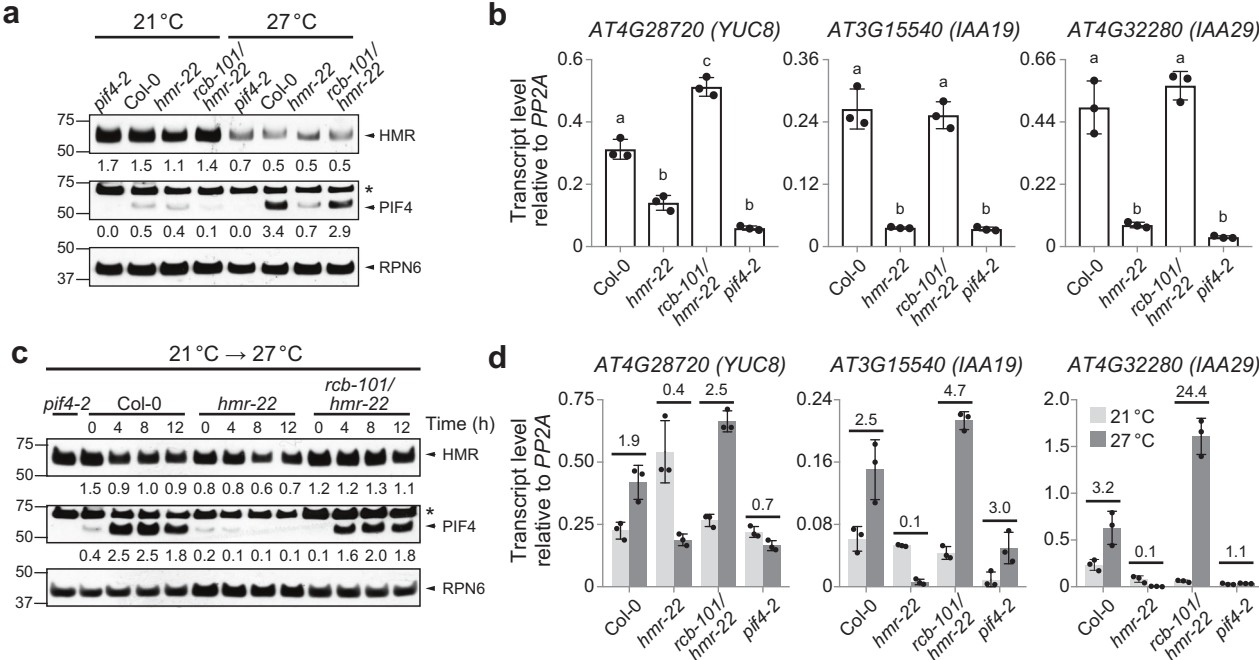

**Fig. 2 *rcb-101/hmr-22* restores PIF4 stability and activity. a** Immunoblot analysis showing HMR and PIF4 levels in Col-0, *hmr-22*, *rcb-101/hmr-22*, and *pif4-2* seedlings grown in 50 µmol m$^{-2}$ s$^{-1}$ R light for 96 h at either 21 or 27 °C. **b** qRT-PCR analysis of the steady-state transcript levels of *YUC8*, *IAA19*, and *IAA29* in Col-0, *hmr-22*, *rcb-101/hmr-22*, and *pif4-2* seedlings grown in 50 µmol m$^{-2}$ s$^{-1}$ R light for 96 h at 27 °C. Different letters denote statistically significant differences in the transcript levels (ANOVA, Tukey's HSD, $p < 0.01$, $n = 3$ biological replicates). **c** Immunoblot analysis of HMR and PIF4 levels in Col-0, *hmr-22*, *rcb-101/hmr-22*, and *pif4-2* seedlings during the 21 to 27 °C transition. Seedlings were grown in 50 µmol m$^{-2}$ s$^{-1}$ R light for 96 h and then transferred to 27 °C in the same light condition. Samples were collected and analyzed at the indicated time points. For (**a**, **c**) RPN6 was used as a loading control. The relative levels of HMR and PIF4, normalized to RPN6, are shown underneath the respective immunoblots. The asterisks indicate non-specific bands. The immunoblot experiments were independently repeated at least three times, and the results of one representative experiment are shown. **d** qRT-PCR analysis of the steady-state transcript levels of *YUC8*, *IAA19*, and *IAA29* in Col-0, *hmr-22*, *rcb-101/hmr-22*, and *pif4-2* during the 21 to 27 °C transition. Seedlings were grown as described in (**c**), samples were taken before (light gray) and 24 h after (dark gray) the 27 °C treatment. Fold changes in the transcript levels after the 27 °C treatment are shown above the columns. For (**b**, **d**), transcript levels were calculated relative to those of *PP2A*. Error bars represent the s.d. of three biological replicates. The centers of the error bars represent the mean values. The source data of the immunoblots in (**a**, **c**) and the qRT-PCR data in (**b**, **d**) are provided in the Source Data file.

plants from moss to flowering plants (Fig. 1e), suggesting an important structural and/or functional role for RCB.

***rcb-101* restores PIF4 stability and activity in *hmr-22*.** HMR regulates daytime thermomorphogenesis by acting as a transcriptional activator to promote PIF4 accumulation as well as the expression of thermoresponsive growth-promoting PIF4 target genes[22]. The *hmr-22* allele dramatically reduces HMR's transactivation activity, leading to severe defects in PIF4 accumulation and the activation of PIF4 target genes[22,51]. Therefore, we examined whether *rcb-101/hmr-22* rescued the defects of *hmr-22* in the stability and activity of PIF4. While PIF4 accumulation was impaired in *hmr-22* at 27 °C, this defect was largely rescued in *rcb-101/hmr-22* (Fig. 2a). Consistent with the rescue of PIF4 accumulation, the expression of three well-characterized thermoresponsive PIF4 target genes involved in auxin biosynthesis and signaling, *YUC8*, *IAA19*, and *IAA29*[22,54], became activated in *rcb-101/hmr-22* at 27 °C (Fig. 2b). Similarly, during the 21 to 27 °C transition, the defects in PIF4 stability and activity in *hmr-22* were also rescued in *rcb-101/hmr-22* (Fig. 2c, d). Together, these results suggest that RCB is another signaling component working in concert with HMR to control PIF4 stability and activity during thermomorphogenesis.

We found that the steady-state level of HMR was lower at 27 °C than at 21 °C (Fig. 2a). However, at both temperatures, the HMR level in *rcb-101/hmr-22* remained similar to those in Col-0 and *hmr-22* (Fig. 2a). Also, the level of HMR did not decrease

significantly in Col-0, *hmr-22*, and *rcb-101/hmr-22* during the 21 to 27 °C transition (Fig. 2c). These results suggest that the phenotypes of *hmr-22* and *rcb-101/hmr-22* are unlikely due to changes in the level of HMR. We have previously shown that photoactivated PHYB enhances the steady-state level of HMR[53]. The decrease in the HMR level at warmer temperatures could be due to a reduction in the overall activity of PHYB in warmer temperatures[6,7] and/or due to temperature-dependent changes in chloroplasts that could potentially impact the stability of chloroplast-localized HMR.

***rcb-101* rescues the defects of *hmr-22* in photomorphogenesis.** Both HMR and RCB are required for PHYB-mediated photomorphogenesis at 21 °C[27,49,51]. Next, we tested whether *rcb-101/hmr-22* could rescue the defects of *hmr-22* in photomorphogenesis. *hmr-22* has a longer hypocotyl under continuous R light at 21 °C, which is due to the accumulation of PIF3 because HMR is required for PIF3 degradation in the light[51]. Interestingly, the long-hypocotyl phenotype of *hmr-22* at 21 °C was rescued in *rcb-101/hmr-22* (Fig. 1a). Consistently, while *hmr-22* failed to degrade PIF3 in R light, this defect was rescued in *rcb-101/hmr-22* (Fig. 3a). We previously reported two classes of HMR-regulated PIF direct target genes: HMR-repressed Class A genes and HMR-induced Class B genes[51]. The misregulation of representative Class A and B genes was either completely or partially rescued in *rcb-101/hmr-22* (Fig. 3b, c). These results, combined with the published data that RCB facilitates PIF3 degradation and

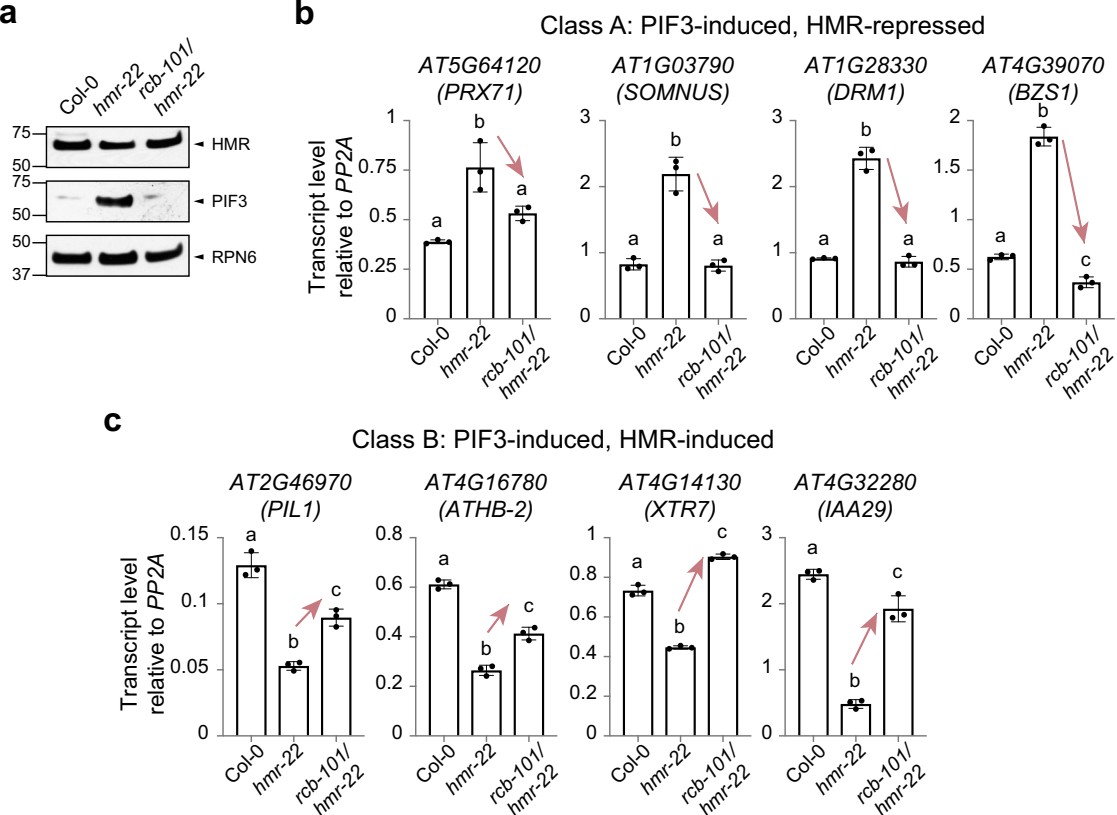

**Fig. 3 *rcb-101* rescues *hmr-22*'s defects in photomorphogenesis. a** Immunoblots showing the levels of HMR and PIF3 in 4-d-old Col-0, *hmr-22*, and *rcb-101/hmr-22* seedlings grown in 10 µmol m$^{-2}$ s$^{-1}$ R light at 21 °C. The immunoblot experiments were independently repeated at least three times, and the results of one representative experiment are shown. **b, c** qRT-PCR analyses of representative class A (**b**) and B (**c**) genes in 4-d-old Col-0, *hmr-22*, and *rcb-101/hmr-22* seedlings grown in 10 µmol m$^{-2}$ s$^{-1}$ R light at 21 °C. Transcript levels were calculated relative to those of *PP2A*. Error bars represent the s.d. of three biological replicates. The centers of the error bars represent the mean values. Different letters denote statistically significant differences in transcript levels (ANOVA, Tukey's HSD, $p < 0.05$, $n = 3$). The source data of the immunoblots in (**a**) and the qRT-PCR data in (**b**, **c**) are provided in the Source Data file.

regulates a similar set of PIF target genes as HMR[27], support the notion that RCB works closely with HMR in the regulation of PIF3 degradation and activity in photomorphogenesis.

***rcb-101* rescues the chloroplast defect of *hmr-22*.** Chloroplast biogenesis is initiated by light via PHY-mediated activation of photosynthesis-associated nuclear-encoded and plastid-encoded genes—*PhANG*s and *PhAPG*s, respectively[14]. In dark-grown seedlings, PIFs, either directly or indirectly, repress the expression of *PhANG*s in the nucleus[55,56]; in parallel, nuclear-localized PIFs also send a yet-unknown nucleus-to-plastid signal to suppress *PhAPG* transcription by blocking the assembly of a 1000-kD plastid-encoded RNA polymerase (PEP) complex in plastids[27,36]. During de-etiolation, light triggers the accumulation of photoactivated PHYs in the nucleus to promote the degradation of PIFs, thereby derepressing chloroplast biogenesis[14]. Therefore, removing PIFs in the nucleus simultaneously lifts the repression of both *PhANG*s and *PhAPG*s[27]. Like HMR, RCB is also dual-targeted to the nucleus and chloroplasts and is required for promoting PEP assembly in the light[27]. However, intriguingly, while HMR plays essential roles in PIF3 degradation in the nucleus and PEP assembly in plastids, RCB controls the activity of the PEP primarily in the nucleus through the degradation of PIFs[27,51].

The weak *hmr-22* allele exhibits pale-green cotyledons and yellowish emerging true leaves due to defects in chlorophyll accumulation (Fig. 4a), but both the cotyledons and true leaves can recover and turn green in later developmental stages[51].

Similar to the *hmr* null alleles, *hmr-22* showed a significant reduction in the expression of PEP-dependent genes, such as *psbB* and *rbcL*, whereas the expression of genes transcribed by the nuclear-encoded plastid RNA polymerase (NEP), such as *rpoB* and *rpoC1*, was elevated (Fig. 4b, c)—a hallmark of mutants deficient specifically in PEP functions[52]. Using antibodies against either a core PEP component, rpoB, or a PEP accessory protein, HMR, we could not detect the 1000-kD PEP complex in *hmr-22* (Fig. 4d), indicating that *hmr-22* is defective in the assembly of the PEP complex. In contrast, *rcb-101/hmr-22* appeared much greener than did *hmr-22* (Fig. 1a). The chlorophyll levels, including those of both chlorophyll a and b, were largely rescued in *rcb-101/hmr-22* (Fig. 4a). Moreover, the expression levels of the PEP- and NEP-dependent marker genes in *rcb-101/hmr-22* were reversed to those in Col-0 (Fig. 4b, c), and PEP assembly was also rescued (Fig. 4d). These results demonstrate that *rcb-101* can completely rescue the chloroplast-biogenesis defects of *hmr-22*. Given our previous findings that RCB controls chloroplast biogenesis primarily by degrading PIFs in the nucleus[27], these results support the idea that *rcb-101* rescues *hmr-22*'s chloroplast defects by restoring PIF3 degradation in the nucleus (Fig. 3a). However, the current data cannot completely exclude the possibility that *rcb-101* also exerts additional effects directly in the plastids.

**RCB is required for thermoresponsive PIF4 accumulation.** The fact that *rcb-101* could rescue the defects of *hmr-22* in both

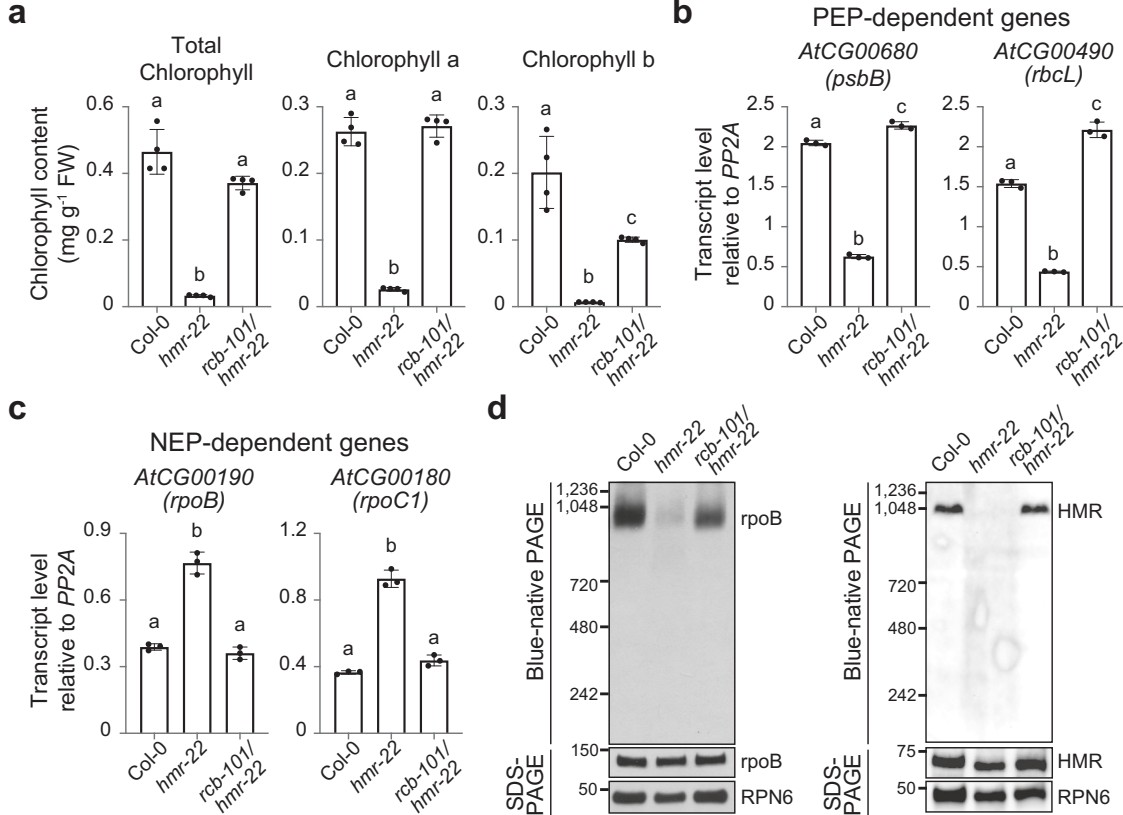

**Fig. 4 *rcb-101* rescues *hmr-22*'s defects in chloroplast biogenesis. a** Quantification of total chlorophyll, chlorophyll a, and chlorophyll b in 4-d-old Col-0, *hmr-22*, and *rcb-101/hmr-22* seedlings grown in 10 μmol m$^{-2}$ s$^{-1}$ R light at 21 °C. **b, c** qRT-PCR analyses of two PEP-dependent genes, *psbB* and *rbcL* (**b**), and two NEP-dependent genes, *rpoB* and *ropC1* (**c**), in 4-d-old Col-0, *hmr-22*, and *rcb-101/hmr-22* seedlings grown in 10 μmol m$^{-2}$ s$^{-1}$ R light at 21 °C. Transcript levels were calculated relative to those of *PP2A*. For **a, b, c**, error bars represent the s.d. of three (**b, c**) or four (**a**) biological replicates. The centers of the error bars represent the mean values. Different letters denote statistically significant differences between the samples (ANOVA, Tukey's HSD, $p < 0.01$, $n = 3$ or 4 biological replicates). **d** Immunoblots showing the level of the PEP complex (blue-native PAGE) and the levels of rpoB (left panel) or HMR (right panel) (SDS-PAGE) in 4-d-old Col-0, *hmr-22*, and *rcb-101/hmr-22* seedlings grown in 10 μmol m$^{-2}$ s$^{-1}$ R light at 21 °C. RPN6 was used as a loading control. The immunoblot experiments were independently repeated at least three times, and the results of one representative experiment are shown. The source data of the chlorophyll measurements in (**a**), the qRT-PCR data in (**b, c**), and the immunoblots in (**d**) are provided in the Source Data file.

photomorphogenesis and thermomorphogenesis suggests that RCB works closely with HMR in PHYB signaling. Next, we tested whether RCB is required for thermomorphogenesis. To that end, we characterized the thermomorphogenetic responses of a null allele, *rcb-10* (Fig. 1c)[27], as well as the *rcb-101* single mutant. Interestingly, *rcb-10* exhibited only 10% of the warm-temperature response of Col-0 (Fig. 5a, b). Similar to the *hmr* mutant, *rcb-10* failed to accumulate PIF4 and is impaired in the activation of PIF4-dependent growth-relevant genes such as *YUC8*, *IAA19*, and *IAA29* at 27 °C (Fig. 5c, d). In contrast, *rcb-101* showed only a slight increase in the relative warm-temperature response (Fig. 5a, b), and it had wild-type-like responses in PIF4 accumulation and the expression of the thermoresponsive PIF4 target genes (Fig. 5e, f). Together, these results demonstrate that RCB is an essential component of PHYB-mediated thermomorphogenesis and that, like HMR, it participates in the regulation of PIF4 stability and activity. The lack of a significant phenotype of the *rcb-101* single mutant suggests that the rescue of *hmr-22* by *rcb-101* might be allele-specific.

**RCB interacts directly with HMR.** To confirm that *rcb-101* is an allele-specific suppressor of *hmr-22*, we crossed *rcb-101* with the null *hmr-5* mutant. The *rcb-101/hmr-5* double mutant showed no significant improvement in the thermal response compared with *hmr-5*, and it remained albino (Fig. 6a, b). These results support

the idea that the effect of RCB$^{A275V}$ (RCB101) relies on the presence of HMR$^{D516N}$ (HMR22), implying a direct interaction between RCB and HMR. To test this hypothesis, we performed reciprocal co-immunoprecipitation (Co-IP) assays using two previously reported transgenic lines, *RCB-HA-His/rcb-10*[27] and *HMR-HA/hmr-5*[51,53], to test a possible interaction between RCB and HMR in vivo. The results of these experiments showed that either HA-tagged RCB or HA-tagged HMR could pulldown the other, indicating that RCB and HMR are associated with each other in vivo (Fig. 6c, d). To further examine whether RCB interacts with HMR directly, we performed in vitro GST pull-down assays using GST-tagged full-length HMR (GST-HMR) and in vitro transcribed and translated HA-tagged RCB (HA-RCB) (Fig. 6e). GST-HMR, but not GST alone, could pulldown HA-RCB (Fig. 6f), confirming that RCB interacts directly with HMR in vitro. HMR contains an N-terminal transit peptide[50], a glutamate-rich region, PHYA-interacting regions 1 and 2 (PIR1 and PIR2)[53], and the C-terminal 9aaTAD[51] (Fig. 6e). To determine which region of HMR confers the interaction with RCB, we tested two N-terminal HMR truncation fragments: GST-HMRΔ115, which contains the glutamate-rich region, PIR2, and 9aaTAD, and GST-HMRΔ251, which contains PIR2 and the 9aaTAD. Only GST-HMRΔ251, but not GST-HMRΔ115, could pulldown HA-RCB (Fig. 6f). These results indicate that the C-terminal half of HMR, including PIR2 and the 9aaTAD, is

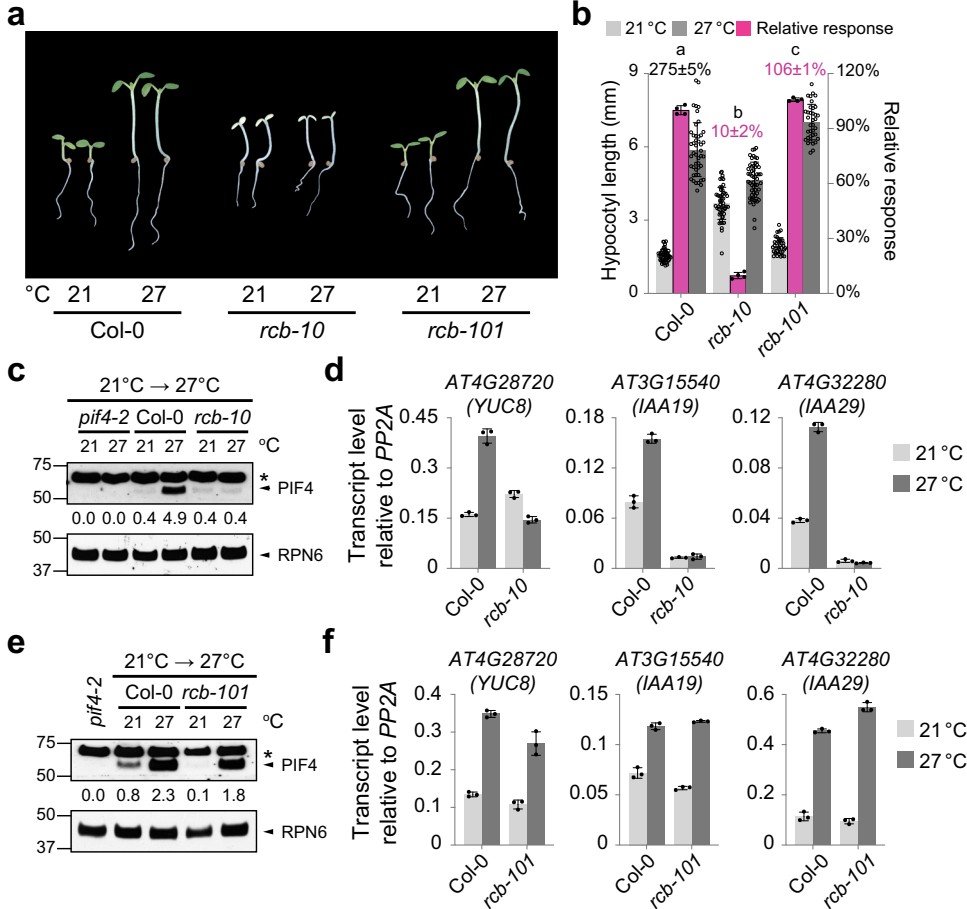

**Fig. 5 RCB is required for thermoresponsive PIF4 accumulation. a** Representative images of 4-d-old Col-0, *rcb-10*, and *rcb-101* seedlings grown in 50 µmol m$^{-2}$ s$^{-1}$ R light at either 21 or 27 °C. **b** Hypocotyl length measurements of the seedlings shown in (**a**). The light- and dark-gray columns represent hypocotyl length measurements in 21 °C and 27 °C, respectively. The percent increase in hypocotyl length (mean ± s.d., $n = 3$ biological replicates) of Col-0 at 27 °C is shown in black above its columns. The magenta columns show the relative response. Error bars for the hypocotyl measurements represent the s.d. ($n > 30$ seedlings); error bars for the relative responses represent the s.d. of four biological replicates. The centers of the error bars represent the mean values. Purple numbers show the mean ± s.d. values of relative responses and different letters denote statistically significant differences in relative responses (ANOVA, Tukey's HSD, $p < 0.01$, $n = 4$ biological replicates). **c** Immunoblot analysis of the PIF4 levels in Col-0 and *rcb-10* during the 21 to 27 °C transition. Samples were collected before (21 °C) and 4 h after (27 °C) the warm-temperature treatment. **d** qRT-PCR analysis of the steady-state transcript levels of *YUC8*, *IAA19*, and *IAA29* in Col-0 and *rcb-10* during the 21 to 27 °C transition. **e** Immunoblot analysis of the PIF4 levels in Col-0 and *rcb-101* during the 21 to 27 °C transition. For (**c**, **e**), seedlings were grown in 50 µmol m$^{-2}$ s$^{-1}$ R light for 96 h and then transferred to 27 °C. RPN6 was used as a loading control. The relative levels of PIF4, normalized to RPN6, are shown underneath the respective immunoblots. The asterisks indicate non-specific bands. The immunoblot experiments were independently repeated at least three times, and the results of one representative experiment are shown. **f** qRT-PCR analysis of the steady-state transcript levels of *YUC8*, *IAA19*, and *IAA29* in Col-0 and *rcb-101* during the 21 to 27 °C transition. For (**d**, **f**), transcript levels were calculated relative to those of *PP2A*. Error bars represent the s.d. of three biological replicates. The centers of the error bars represent the mean values. The source data of the hypocotyl measurements in (**b**), the immunoblots (**c**, **e**), and the qRT-PCR data in (**d**, **f**) are provided in the Source Data file.

sufficient to mediate the interaction with RCB. The fact that GST-HMRΔ115 did not interact with HA-RCB could be due to either improper folding of the RCB-interacting region in this recombinant protein or a potential inhibitory role of the middle region (between amino acids 116 and 252) to the HMR-RCB interaction. To examine which region of RCB is involved in the HMR interaction, we used GST-HMRΔ251 as the bait to pulldown three in vitro translated, HA-tagged truncation fragments of RCB (Fig. 6e): HA-RCBΔ51, which lacks the transit peptide, mimicking the endogenous RCB[27]; HA-RCBΔ89; and HA-RCBΔ159, which contains only the Trx fold. The results showed that GST-HMRΔ251 could pulldown all three RCB fragments with similar affinities (Fig. 6g). However, comparing the interaction with that with the full-length RCB, we consistently detected a decrease in the affinity between GST-HMRΔ251 and the RCB truncation fragments, suggesting that the transit peptide of RCB contributed

significantly to the binding between full-length RCB and HMR (Fig. 6f, g). However, we have previously shown that the mature forms of HMR and RCB (without the transit peptide) are present in both plastids and the nucleus[27,50]. These results thus suggest that the interaction between the mature forms of HMR and RCB is mediated by their respective C-terminal halves, which include the two domains harboring mutations in the *rcb-101* and *hmr-22*—the 9aaTAD in HMR (which carries the D516N mutation in HMR22) and the Trx fold in RCB (which carries the A275V mutation in RCB101). An obvious hypothesis would be that RCB101 rescues a defect in the interaction between HMR22 and the wild-type RCB. However, our experiments did not detect any observable changes in the HMR-RCB interaction caused by the D516N mutation in HMR22 (Supplementary Fig. 2). These negative results could be due to the limited resolution of our pulldown assays to detect subtle differences. Alternatively, it is

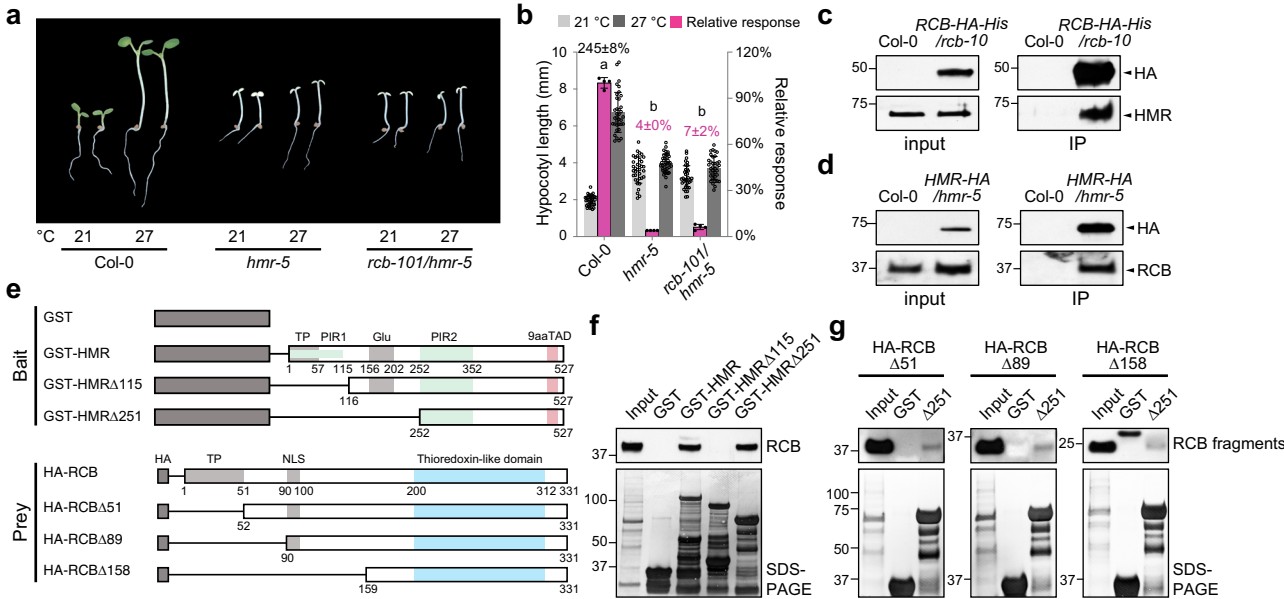

**Fig. 6 RCB directly interacts with HMR. a** Representative images of 4-d-old Col-0, *hmr-5*, and *rcb-101/hmr-5* seedlings grown in 50 µmol m$^{-2}$ s$^{-1}$ R light in either 21 or 27 °C. **b** Hypocotyl length measurements of the seedlings shown in (**a**). The percent increase in hypocotyl length (mean ± s.d., $n = 4$ biological replicates) of Col-0 in 27 °C is shown in black above its columns. The magenta bars show the relative response. Error bars for the hypocotyl measurements represent s.d. ($n > 30$ seedlings); error bars for the relative responses represent the s.d. of four biological replicates. The centers of the error bars represent the mean values. Purple numbers show the mean ± s.d. values of relative responses and different letters denote statistically significant differences in relative responses (ANOVA, Tukey's HSD, $p < 0.01$, $n = 4$ biological replicates). **c, d** Reciprocal co-immunoprecipitation experiments showing the interaction between RCB and HMR in vivo. **c** Immunoblots showing the co-immunoprecipitation results using 4-d-old *RCB-HA-His/rcb-10* seedlings grown in 10 µmol m$^{-2}$ s$^{-1}$ R light. RCB-HA-His was pulled-down using Pierce™ anti-HA agarose. RCB-HA-His and HMR from the input and pulldown fractions were detected by immunoblots using anti-HA and anti-HMR antibodies, respectively. **d** Immunoblots showing the co-immunoprecipitation results using 4-d-old *HMR-HA/hmr-5* seedlings grown in 10 µmol m$^{-2}$ s$^{-1}$ R light. HMR-HA was pulled-down using Pierce™ anti-HA agarose. HMR-HA and RCB from the input and pulldown fractions were detected by immunoblots using anti-HA and anti-RCB antibodies, respectively. For (**c, d**), samples from Col-0 seedlings grown in the same conditions were used as negative controls. **e** Schematics of the bait and prey proteins used in the GST pulldown assays to show a direct interaction between HMR and RCB. TP: transit peptide; Glu: Glu-rich region; PIR2: PHYA-interacting region 2; 9aaTAD: 9-amino-acid TAD (amino acids 512-520). **f** GST pulldown assays using GST-tagged full-length or N-terminal truncations to pulldown in vitro translated HA-tagged full-length RCB (HA-RCB). **g** GST pulldown assays using GST-tagged HMRΔ251 (Δ251) to pulldown in vitro translated HA-tagged N-terminal RCB truncations. For (**f, g**), the upper panels are immunoblots using anti-HA antibodies showing the bound and input fractions of either HA-tagged full-length RCB (**f**) or the various RCB truncation fragments (**g**); the lower panels are Coomassie Blue-stained SDS-PAGE gels showing immobilized GST and GST-tagged HMR and HMR fragments. The pulldown experiments in (**c, d, f, g**), were independently repeated two times, and the results of one representative experiment are shown. The source data of the hypocotyl measurements in (**b**) and the immunoblots in (**c, d, f, g**) are provided in the Source Data file.

also possible that these mutations influence the activities of HMR and RCB without perturbing their interaction.

## Discussion

Plants are more likely to encounter warmer temperatures in the natural environment during the daytime—a condition when they simultaneously sense light and temperature cues. However, the molecular mechanism of daytime temperature signaling remains poorly understood. Thermomorphogenesis during both the day-time and nighttime in *Arabidopsis* is triggered by warm-temperature-induced accumulation of the central thermo-regulator PIF4[7,16,20,22]. One clear distinction in the daytime is that PIF4 needs to be particularly stabilized to avoid degradation promoted by active PHYB (Fig. 7). We have previously shown that PIF4 stabilization in the light requires a PIF4-interacting transcriptional activator, HMR[22]. In this study, using a forward genetic screen for suppressors of the *hmr-22* allele, we identified RCB as another missing piece in daytime temperature signaling for the control of PIF4 stabilization. The combined genetics and biochemical evidence draws a novel link between RCB and HMR in temperature signaling and strongly supports the model that RCB works collaboratively with HMR to initiate thermo-morphogenesis by stabilizing PIF4 in the daytime (Fig. 7).

The regulation of the stability of PIFs, including PIF3 and PIF4, is at the center of PHYB signaling. PIF3 and PIF4 (as well as PIF1 and PIF5) are rapidly degraded when dark-grown seedlings are exposed to light, which is considered the central mechanism for de-etiolation[37,39,40]. The current model posits that PHYB inter-acts directly with PIF3 and PIF4 to trigger their phosphorylation and ubiquitin–proteasome-dependent proteolysis[18,39,57]. PIF3 and PIF4 accumulate to high levels in the *phyB* mutant in the light[22], indicating that PHYB-mediated degradation mechanisms for PIF3 and PIF4 operate prominently during the daytime. Consistently, PIF3 does not accumulate to a detectable level in continuous light under ambient growth temperatures[58–60]. However, in striking contrast, PIF4 can accumulate to significant levels in the light and is stabilized particularly by elevated tem-peratures to initiate thermomorphogenesis[22,47], implying that a unique warm-temperature-dependent mechanism exists to insu-late PIF4 from PHYB-mediated degradation. In addition, PIF7, whose stability is light-independent[41], also accumulates to higher levels in warm temperatures and plays an equally important, nonredundant role from PIF4 in thermomorphogenesis[45]. Accumulating evidence supports an emerging theme that differ-ent temperature ranges engage with PHYB signaling by mod-ulating the stability of distinct PIFs. For example, under cold

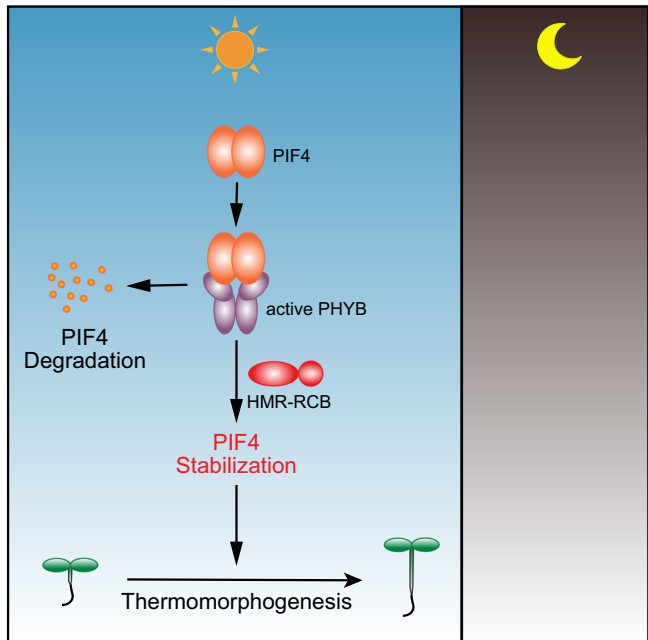

**Fig. 7 Model for daytime PIF4 stabilization in thermomorphogenesis.** During the daytime under LD conditions, PIF4 interacts with active PHYB and is promoted for degradation. HMR and RCB collaboratively enable thermomorphogenesis by stabilizing PIF4.

temperatures, the level of PIF3 (but not PIF4) becomes elevated in the daytime to modulate freezing tolerance[61,62]. The fact that different PIFs are utilized to modulate the responses by distinct temperature ranges suggests that individual PIFs govern distinct nodes in the signaling networks, allowing the integration of light and various temperature cues. Individual PIFs must confer discernable activities to regulate a diverse downstream response, which could be achieved in part by the specificity of their direct target genes[63]. One mechanism underpinning the stabilization of individual PIFs is that different PIFs are degraded via distinct mechanisms. For instance, PIF3 degradation is mediated by the Cullin3-LRB [(Light-Response Bric-a-Brack/Tramtrack/Broad (BTB)] and Cullin1-EBF (EIN3-Binding F Box Protein) E3 ubiquitin ligases[64,65], whereas PIF4 degradation is mediated by Cullin3-based E3 ubiquitin ligases with BLADE-ON-PETIOLE (BOP) 1 and 2 as the substrate recognition subunit[66]. The discovery of HMR and RCB suggest that specific PHYB signaling components enable selective regulation of individual PIFs under unique combinations of light and temperature conditions. Both HMR and RCB play essential roles in PIF3 degradation in continuous light. Similar to *phyB-9*, in *hmr-5* and *rcb-10*, PIF3 degradation is blocked in the light[27,49], indicating that HMR and RCB mechanistically participate in PHYB-mediated PIF3 degradation. However, HMR acts antagonistically to PHYB signaling by stabilizing PIF4 in thermomorphogenesis[22]. The present results, combined with those of our previous studies[27], demonstrate that RCB acts similarly to HMR by playing opposing roles in promoting PIF3 degradation and PIF4 stabilization (Figs. 2, 5). Together, the current results suggest that HMR and RCB are critical components of the mechanism that allows selective stabilization of individual members of the PIF transcription factor family for eliciting specific responses under a unique combination of light and temperature conditions, e.g., thermomorphogenesis in warmer long days (Fig. 7).

Warm temperature usually occurs coinciding with high light. However, based on principle, the warm-temperature-dependent

effect on PHYB should be largest at low light intensities[6]. In this study, we show that warm temperatures caused an even greater difference in hypocotyl length between 21 and 27 °C in 50 μmol m$^{-2}$ s$^{-1}$ R light compared with the previously used condition of 10 μmol m$^{-2}$ s$^{-1}$ R light (Fig. 1)[22]. More importantly, the antagonistic action by warm temperature on PHYB-dependent PIF4 degradation operates effectively in relatively high light intensities (Fig. 2a). Thus, these results, combined with the previous studies[22,45], provide experimental evidence supporting that warm temperature can effectively modulate PHYB signaling in high light conditions.

Both PIF3 degradation and PIF4 stabilization rely on the activity of HMR's 9aaTAD, suggesting a causal relationship between the transcriptional activity of HMR and the stability of PIF3 and PIF4[22,51]. The *hmr-22* allele, which carries a D516N mutation in HMR's 9aaTAD and reduces its transactivation activity by about 60% in yeast, is impaired in the activation of a specific set of PIF target genes as well as PIF3 degradation and PIF4 stabilization[22,51]. A fusion between the TAD of the herpes simplex virus protein VP16 and HMR22 (HMR22-VP16) could rescue the activation of PIF3 target genes; however, HMR22-VP16 was unable to rescue PIF3 degradation[51]. Based on these results, we proposed that the activation of a subset of PIF3 target genes—the Class B genes, including *PIL1*, *IAA29*, *ATHB-2*, and *XTR7*—in the light relies on HMR's 9aaTAD and is coupled with HMR-dependent PIF3 degradation[51]. The fact that HMR22-VP16 was unable to restore PIF3 degradation implies that PIF3 degradation may require a unique function or a special transactivation mechanism of HMR's 9aaTAD. In thermomorphogenesis, HMR22-VP16 can restore the expression of thermoresponsive PIF4 target genes and stabilize PIF4 in *hmr-22*, suggesting that PIF4 stabilization depends on an event during or after the transcriptional activation of PIF4 target genes[22]. Here we demonstrate that *rcb-101* rescues all examined *hmr-22* defects, including PIF3 degradation (Fig. 3a), the expression of the Class B genes (Fig. 3c), PIF4 stabilization (Fig. 2a, c), and the activation of thermoresponsive PIF4 targets (Fig. 2b, d). Moreover, RCB interacts directly with HMR (Fig. 6). Together, these results support a new hypothesis that RCB interacts with HMR and participates in HMR's transcriptional activation activity. In this scenario, removing RCB or HMR would lead to similar effects. Consistent with this prediction, *rcb-10* shares many similarities with *hmr-5*, including similar phenotypes at 21 and 27 °C, a largely overlapping misregulated genes, the same defects in PIF3 degradation and PIF4 stabilization[27,49,51,53]. Moreover, the *rcb-10/hmr-5* double mutant exhibits similar phenotypes as the *rcb-10* and *hmr-5* single mutants[27].

Studies in yeast and animal models have shown an intimate relationship between the stability and activity of transcriptional activators. Ubiquitylation and subsequent proteasome-mediated degradation of activators could be an integral part of transcriptional activation[67,68]. For example, the transcriptional activity of VP16 in yeast requires its E3 ubiquitin ligase Met30[69]. In the absence of Met30, VP16 is stabilized but not transcriptionally active[69]—a scenario mimicking the accumulation of inactive PIF3 in the *hmr* and *rcb* mutants[49,51]. Similarly, the transcriptional activity of the proto-oncogene product Myc in human cells depends on Skp2, the substrate recognition subunit of a Cullin-based E3 ubiquitin ligase for Myc degradation[70]. However, the mechanism underlying the transcription-coupled degradation of transcription factors is still not fully understood. One possible mechanism is that the transcription activation domains of VP16 and Myc are required for recruiting the E3 ubiquitin ligases for their degradation. Alternatively, the ubiquitylation of activators is required for recruiting the proteasome, which has been suggested to play an important role in transcription activation, in addition to its conventional role in protein degradation[71]. The stabilization

of transcription factors has also been linked to transcriptional activity. For example, during hematopoietic cell differentiation, the interaction between two transcriptional regulators, SCL and LMO2, stabilizes LMO2 by preventing its ubiquitin–proteasome-dependent degradation[72]. Unlike these published examples, the stability of PIF3 and PIF4 is controlled by the TAD of a separate activator, HMR, and an HMR-interacting protein, RCB; additionally, the same TAD of HMR can selectively mediate either degradation or stabilization of PIF3 and PIF4, respectively[22,51]. The accumulation of PIF4 (and PIF5) in the light also depends on MYB30, an R2R3-MYB family transcription factor that interacts directly with PHYs and PIFs[44], lending further support to a link between the stability of PIFs and their transcriptional activities. The transcriptional activity of PIF4 in thermomorphogenesis is negatively regulated by two transcriptional regulators and components of the circadian clock, TIMING OF CAB EXPRESSION1 (TOC1) and PSEUDO-RESPONSE REGULATOR5 (PRR5)[21]. However, it remains unclear as to whether TOC1 and PRR5 can also influence the stability of PIF4. Moreover, PIF4 accumulation relies on two antagonists of PHYB signaling, DE-ETIOLATED1 (DET1) and CONSTITUTIVE PHOTOMORPHOGENIC1 (COP1)[20,47,73]. Unlike HMR and RCB, DET1 and COP1 are also required for PIF3 stabilization; their links to HMR and RCB remain unknown.

Both RCB and HMR interact directly with PHYB and are required for the subnuclear localization of PHYB to PHYB-containing subnuclear foci called photobodies[27,49,74]. Photobody formation is dynamically regulated by both light and temperature[12,75,76], and is largely impaired in the hmr and rcb mutants[27,49,53]. Photobody localization of PHYB is tightly correlated with PIF3 degradation[76]. Although it is still unclear whether photobodies play a role in PIF4 stabilization, the fact that both the hmr and rcb mutants, which are defective in photobody formation[27,49,53], also impede PIF4 stabilization and thermomorphogenesis (Fig. 5c)[22], implies a role for photobodies in PIF4 stabilization. Therefore, it is also possible that HMR and RCB control PIF3 degradation and PIF4 stabilization through the regulation of the formation of photobodies.

RCB and HMR are dual-targeted to the nucleus and plastids (chloroplasts), playing central roles in the nucleus-to-plastid or anterograde signaling pathway that controls the light-dependent assembly of the PEP for the transcription of *PhAPGs*[27,49,50,52]. HMR regulates plastid transcription through two distinct nuclear and plastidic functions[51]. While nuclear HMR mediates the degradation of the master chloroplast repressors, PIFs, plastidic HMR is an accessory protein of the PEP complex and is required for its assembly[27,49,52]. We have previously shown that *hmr-22* is impaired in PIF1 and PIF3 degradation in the light[51]. Our results here demonstrate that PEP assembly, as well as the expression of *PhAPGs*, are severely impaired in *hmr-22* (Fig. 4), explaining the pale-green phenotype of *hmr-22*. It was unclear whether the chloroplast defects in *hmr-22* were caused directly by a defect of HMR22's function in PEP assembly or indirectly through the accumulation of PIFs in the nucleus. The results here show that *rcb-101/hmr-22* restored PIF3 degradation (Fig. 3a) and rescued all the defects related to chloroplast biogenesis (Fig. 4), supporting the idea that *rcb-101* rescues the chloroplast defects of *hmr-22* in the nucleus by removing PIF3. Because *rcb-101/hmr-22* restored PIF4 accumulation, these results also suggest that PIF4 plays a minor role in repressing chloroplast biogenesis. RCB, also called SVR4 (SUPPRESSOR OF VARIEGATION 4)/ MRL7 (MESOPHYLL-CELL RNAi LIBRARY LINE 7)/AtECB1 (ARABIDOPSIS EARLY CHLOROPLAST BIOGENESIS 1), has been shown to perform important functions in plastids. For example, RCB is associated with the PEP complex in the nucleoid and is required for the maintenance of the photosynthetic

apparatus[77,78]. Therefore, our results could not completely rule out the possibility that the rescue of *hmr-22*'s chloroplast defects by *rcb-101* also involves RCB's functions in plastids.

In conclusion, this study uncovers RCB as an essential component of temperature signaling in *Arabidopsis* thermomorphogenesis. Our genetic and biochemical results draw a direct link between RCB and HMR in early temperature signaling, highlighting their collaborative role in the stabilization of PIF4 during the daytime. Our results support the emerging theme that daytime temperature signaling relies on temperature-dependent mechanisms to stabilize individual PIFs. Future studies will be aimed at understanding the molecular and cellular mechanisms by which the stability and activity of individual PIFs are selectively controlled by diverse combinations of light and temperature environments.

## Methods

**Plant materials and growth conditions**. The *Arabidopsis* mutants *pif4-2* (SAIL_1288_E07) and *rcb-10* (SALK_075057) were previously described[25,27] and obtained from the *Arabidopsis* Biological Resource Center. The *hmr-5* and *hmr-22* mutants, as well as the *HMR-HA/hmr-5* and *RCB-HA-His/rcb-10* transgenic lines, were previously described[22,27,51,53]. *hmr-1* in the Ler background was generated by backcrossing *hmr-1/PBG*[49] to Ler. Seeds were surface sterilized with 70% ethanol and bleach and plated on half-strength Murashige and Skoog (1/2 MS) media supplemented with Gamborg's vitamins (MSP0506, Caisson Laboratories, North Logan, UT), 0.5 mM MES (pH 5.7), and 0.8% (w/v) agar (A038, Caisson Laboratories, North Logan, UT)[51]. Seeds were stratified in the dark at 4 °C for 5 days before treatment under specific light and temperature conditions in LED chambers (Percival Scientific, Perry, IA). Fluence rates of light were measured using an Apogee PS200 spectroradiometer (Apogee Instruments Inc. Logan, UT).

**Hypocotyl measurement**. Seedlings were treated with specific light and temperature conditions for 96 h. At least thirty seedlings from each sample were scanned using an Epson Perfection V700 photo scanner, and hypocotyl length was measured using NIH ImageJ software (http://rsb.info.nih.gov/nih-image/). The percent increase (PI) in the hypocotyl length of each line was calculated as the percentage of the increase in hypocotyl length at 27 °C relative to that at 21 °C. The relative response of a mutant is defined as the percentage of its PI value or temperature response relative to that of Col-0. At least three replicates were used to calculate the mean and standard deviation of each relative response. Bar charts were generated using Prism 8 (GraphPad Software, San Diego, CA).

**EMS mutagenesis and *hmr-22* suppressor screen**. *hmr-22* seeds were mutagenized with EMS. First, 0.2 g *hmr-22* seeds were hydrated in 45 ml of ddH$_2$O with 0.005% Tween-20 for 4 h, washed with ddH$_2$O twice, and then soaked in 0.2% EMS (MilliporeSigma, St. Louis, MO) for 15 h with rotation. Subsequently, the seeds were washed with ddH$_2$O 8 times, stratified in the dark at 4 °C for 4 days, and sown onto 1/2 MS plates. A total of 1920 M1 seedlings were randomly selected and grown to flowering. M2 seeds were collected from each M1 plant individually. We then performed family screening for *hmr-22* suppressors using the M2 seeds from the 1920 families. At least eighty M2 seeds from each M1 family were grown in 50 μmol m$^{-2}$ s$^{-1}$ continuous R light for 4 days at 27 °C, and putative suppressors with a significantly longer hypocotyl than *hmr-22* were kept and subjected to a secondary screen for the hypocotyl phenotype at 21 °C to confirm that the warm-temperature response has been rescued.

**RNA extraction and qRT-PCR**. Seedlings (50–100 mg) were collected by flash freezing in liquid nitrogen and stored at −80 °C until processing. Samples were ground to a fine powder in liquid nitrogen, and RNA was extracted using a Quick-RNA MiniPrep kit with on-column DNase I digestion (Zymo Research, Irvine, CA). cDNA synthesis was performed with 2–2.5 μg total RNA using a Superscript II First Strand cDNA Synthesis Kit (Thermo Fisher Scientific, Waltham, MA). For qRT-PCR, cDNA diluted in nuclease-free water was mixed with iQ SYBR Green Supermix (Bio-Rad Laboratories, Hercules, CA) and primers (Supplementary Table 2). qRT-PCR reactions were performed in triplicate with a Bio-Rad CFX Connect Real-Time PCR Detection System. Transcript levels were calculated relative to the level of PP2A. Bar charts were generated using Prism 8 (GraphPad Software, San Diego, CA).

**Protein extraction and immunoblots**. Seedlings (100–250 mg) were harvested and directly homogenized using a Mini-Beadbeater-24 (BioSpec Products, Bartlesville, OK) in three volumes (mg/μL) of extraction buffer containing 100 mM Tris-Cl pH 7.5, 100 mM NaCl, 5 mM EDTA, 5% SDS, 20% glycerol, 20 mM DTT, 40 mM β-mercaptoethanol, 2 mM phenylmethylsulfonyl fluoride, 40 μM MG115, 40 μM MG132, 10 mM N-ethylmaleimide, 1× phosphatase inhibitor

cocktail 3 (MilliporeSigma, Burlington, MA), 1× EDTA-free protease inhibitor cocktail (MilliporeSigma, Burlington, MA), and 0.01% bromophenol blue. Samples were immediately boiled for 10 min and centrifuged at 16,000 × g for 10 mins. Protein samples in the supernatant were stored at −80 °C or used immediately for immunoblots.

Cleared protein samples were separated via SDS-PAGE, transferred to nitrocellulose membranes, probed with the indicated primary antibodies, and then incubated with 1:5000 dilution of horseradish peroxidase-conjugated goat anti-rabbit or anti-mouse secondary antibodies (Bio-Rad Laboratories, 1706515 for anti-rabbit and 1706516 for anti-mouse). Primary antibodies, including monoclonal mouse anti-HA antibodies (MilliporeSigma, H3663), polyclonal rabbit anti-HMR antibodies[49], polyclonal rabbit anti-RCB antibodies[27], polyclonal rabbit anti-PIF4 antibodies (Agrisera, AS12 1860), monoclonal mouse anti-rpoB antibodies (PhytoAB, PHY1700), and polyclonal rabbit anti-RPN6 antibodies (Enzo Life Sciences, BML-PW8370-0100) were used at 1:1000 dilution. Signals were detected via chemiluminescence using a SuperSignal kit (Thermo Fisher Scientific).

**Co-immunoprecipitation.** Light-grown seedlings grown in 21 °C (500 mg to 1 g) were harvested, flash-frozen in liquid nitrogen, and stored at −80 °C until processing. Samples were ground to a fine powder in liquid nitrogen and homogenized in two volumes (mg/μL) of Co-IP buffer containing 50 mM Tris-Cl pH 7.5, 100 mM NaCl, 1 mM EDTA, 2 mM DTT, 0.1% NP-40, 1× EDTA-free protease inhibitor cocktail (MilliporeSigma), 1 mM phenylmethylsulfonyl fluoride, 40 μM MG115, 40 μM MG132, and 10 mM N-ethylmaleimide. After clearing with two rounds of centrifugation at 20,000 × g for 10 min at 4 °C, the lysate was incubated with 50 μl equilibrated anti-HA affinity matrix (Roche) for 4 h at 4 °C. Following the incubation, the matrix was washed 4 times with 1 ml Co-IP buffer, and protein samples were eluted by boiling in 100 μl 2× SDS loading buffer containing 100 mM Tris-Cl, pH 6.8, 4% SDS, 12% glycerol, 40 mM β-mercaptoethanol, 200 mM DTT, and 0.01% bromophenol blue. Thirty microliters of eluates were used in subsequent SDS-PAGE and immunoblotting.

**GST pulldown.** GST pulldown assays were performed as described previously[51]. Full-length or truncated CDS of *HMR* were cloned into pET42b vectors (Supplementary Table 3) and expressed as GST-HMR fusion proteins in the *E. coli* strain BL21 (DE3) (Agilent Technologies). Full-length or truncated CDS of *RCB* were cloned into pCMX-PL2-NterHA vectors (Supplementary Table 3) and expressed as HA-RCB proteins using the TNT T7 Coupled Reticulocyte Lysate System (Promega). HA-RCB prey proteins were incubated with the affinity-purified GST-HMR bait proteins immobilized on glutathione Sepharose beads (GE Healthcare) at 4 °C for 2 h. Beads were washed four times with E buffer (50 mM Tris-HCl, pH 7.5, 100 mM NaCl, 1 mM EDTA, 1 mM EGTA, 1% DMSO, 2 mM DTT, 0.1% Nonidet P-40). Bound proteins were eluted by boiling in 2× SDS loading buffer and used in subsequent SDS-PAGE and immunoblotting. Input and immunoprecipitated HA-RCB prey proteins were detected using goat anti-HA polyclonal antibodies (GenScript). The amount of GST-HMR bait proteins was visualized by staining the SDS-PAGE with Coomassie Brilliant Blue.

**Blue-native gel electrophoresis.** The status of the PEP complex assembly was analyzed via blue-native polyacrylamide gel electrophoresis (BN-PAGE) and immunoblot analyses. Seedlings (100 mg) were harvested and flash-frozen in liquid nitrogen, ground to a fine powder, and resuspended in three volumes (mg/μL) of BN lysis buffer containing 100 mM Tris-Cl pH 7.5, 10 mM MgCl$_2$, 25% glycerol, 1% Triton X-100, 10 mM NaF, 5 mM β-mercaptoethanol, and 1× EDTA-free protease inhibitor cocktail (MilliporeSigma). Protein extracts were mixed with the BN sample buffer containing 1× NativePAGE sample buffer, 50 mM 6-aminocaproic acid, 1% n-dodecyl β-D-maltoside (DDM), and Benzonase nuclease using a NativePAGE Sample Prep Kit (Thermo Fisher Scientific). After incubation for 1 h at room temperature, BN-PAGE protein samples were mixed with 0.25% NativePAGE Coomassie blue G-250 sample additive and centrifuged at 17,500 × g for 10 min at 4 °C. Protein samples in the supernatant were separated on 4–16% NativePAGE Bis-Tris protein gel at a constant 30–40 V for 3 h at 4 °C with Dark Blue Cathode Buffer until the blue dye migrated through one-third of the gel and further separated at a constant 20–25 V overnight at 4 °C with Light Blue Cathode Buffer. The separated proteins were transferred onto a polyvinylidene difluoride membrane at a constant 70 V for 7 h at 4 °C. The membrane was destained with methanol for 3 min, probed with the indicated primary antibodies, and incubated with the indicated secondary antibodies mentioned above.

**Chlorophyll measurement.** Total chlorophyll from about 100 mg of seedlings of the indicated genotypes and growth conditions was extracted in 3 ml of 100% DMSO with incubation at 65 °C for 30 min. The absorbances at 665 and 648 nm of 1 ml extract were measured by spectrophotometry. The concentrations (mg/g fresh weight) of total chlorophyll, chlorophyll a, and chlorophyll b were quantified using the equations below as previously described[79]. Total chlorophyll = $(7.49 × OD_{665} + 20.34 × OD_{648}) × 1$ ml/fresh weight; chlorophyll $a = (14.85 × OD_{665} − 5.14 × OD_{648}) × 1$ ml/fresh weight; chlorophyll $b = (25.48 × OD_{648} − 7.36 × OD_{665}) × 1$ ml/fresh weight. Four replicates were used to calculate the mean and standard deviation of each genotype. Bar charts were generated using Prism 8 (GraphPad Software).

**Reporting summary.** Further information on research design is available in the Nature Research Reporting Summary linked to this article.

## Data availability

*Arabidopsis* mutants and plasmids generated during the current study are available from the corresponding authors on reasonable request. Source data are provided with this paper.

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

## Acknowledgements

This work was supported by National Institute of General Medical Sciences grants R01GM132765 and R01GM087388 to M.C., startup funds to Y.Q. from the University of Mississippi (Oxford, MS), and funds of the Hatch Project CA-R-BPS-5186-H to M.C. from the National Institute of Food and Agriculture.

## Author contributions

Y.Q., E.K.P., C.Y.Y., and M.C. conceived the original research plan; M.C. supervised the experiments; Y.Q., E.K.P., C.Y.Y., J.H., H.W., A.B., M.L., H.L., and S.C. performed the experiments; Y.Q., E.K.P., C.Y.Y., J.H., H.W., M.L., H.L., S.C., and M.C. analyzed the data; Y.Q., E.K.P., C.Y.Y., J.H., and M.C. wrote the article with contributions from all authors.

## Competing interests

The authors declare no competing interests.
