## [Peer Review File · Nature Communications]

REVIEWER COMMENTS

Reviewer #1 (Remarks to the Author):

This latest paper by the Meng lab extends our understanding of an important mechanism that controls thermomorphogenesis in the model species *Arabidopsis*. The study complements a recent publication from the Meng lab which reported the discovery of REGULATOR OF CHLOROPLAST BIOGENESIS (RCB) as a nuclear/plastidial phytochrome signaling component required for plastid-encoded RNA polymerase (PEP) complex assembly (Yoo et al., *Nature Communications* 2019 10: 2629). This new submission describes the identification of a *rcb-101* allele as a dominant suppressor of *hmr-22* in thermo-regulation and proceeds to demonstrate that RCB operates with HEMERA (HMR), to control PIF3 signaling, and PIF4-mediated thermomorphogenesis. I feel that the data presented is of an excellent quality and largely supports the hypotheses proposed. I would like to offer the following comments:

The introductory and follow-on narrative could be tightened up a little. The authors firmly bring the reader's attention to the fact that phyB acts as a thermosensor (also final figure diagram), but it is difficult to link the potential loss of phyB(Pfr) activity at higher temperatures, with HMR-RCB module activity in thermomorphogenesis.

The authors give the impression that PIF4 expression peaks at different times of day in SD and LD (an example from p6 is provided below*), but in both LD and SD, PIF4 expression peaks at a similar time - around ZT8 (see Mockler datasets, <http://diurnal.mocklerlab.org/>).

It is (as the authors state) well established that in SDs, PIF4 transcript levels rise to higher levels during the long nights, than in LD. Further, because of this and as phyB activation promotes PIF4 degradation post-dawn, PIF4 protein is more abundant prior to dawn in SDs. This should be made clearer in the text.

Also evident is that in SDs (as for LDs), PIF4 protein levels remain abundant during the light period (Yamashino et al., *Plant Signal Behav* 2013 Mar;8(3):e23390). The manuscript gives the impression that this is not the case.

Page 5 states "PHYB controls seedling morphogenesis primarily by regulating the stability of a family of basic helix-loop-helix transcription factors called PHYTOCHROME-INTERACTING FACTORS (PIFs)". phyB is also known to regulate PIF activity through a sequestering mechanism - relevant to this study, this regulatory mechanism is proposed to be particularly important during the photoperiod (Park et al., *Plant Cell*. 2018 Jun;30(6):1277-1292).

Page 6 states "Notably, in the SD scenario, PIF4 expression is induced when PHYB has reverted to the inactive Pr. Therefore, the regulation of PIF4 stability by PHYB is not a major mechanism in thermomorphogenesis under SD conditions." Could this be explained a bit more fully.

*On page 6 the authors state: "because PIF4 transcripts peak during the daytime in LD, PIF4 transcription is negatively regulated by the transcriptional regulator ELONGATED HYPOCOTYL 5 (HY5)45". This could give the impression that HY5 just operates in LDs, however, to my knowledge the *hy5* mutant is effective in both SD, 12:12 and LD, implying HY5 operates across photoperiods (e.g. Ang and Deng, *Plant Cell* 1994 6, 613-628; Toledo-Ortiz *Plos Genet* 2014, 10;6, e1004416; Gangappa and Kumar *Cell Rep*. 2017 18(2):344-351).

Page 6 states "under LD, in striking contrast to SD conditions, because PIF4 needs to accumulate in the light when PHYB is in the active Pfr, a critical mechanism must be implemented to stabilize PIF4 or antagonize PHYB-mediated PIF4 degradation". Maybe a slight rewording required since the authors have already suggested that phyB is proposed to be less active in warm conditions, (so maybe there is less of a requirement to "antagonize" phyB action).

Comments on the data.

Fig 2a shows HMR protein levels are reduced after 96h at 27°C compared to 21°C. As the authors suggest, this is consistent with expected increase in phyB thermal reversion at the warmer 27°C temperature. The lab's published data showed phyBPfr promotes HMR protein accumulation (Rafaelo 2012), so increased thermal reversion may reduce steady state Pfr levels, and lead to a reduction in HMR. Despite the lower HMR levels in 27°C compared to 21°C, HMR (and RCB) appears to be important for controlling PIF4 protein levels at 27°C and not 21°C (Fig 2a). This data is slightly counterintuitive and may indicative of another temperature-regulated mechanism at play. Another thing to consider is whether the temperature shift to 27°C is sufficient to reduce phyBPfr:Pr steady state levels at the red fluence rates used (50 μmol). Perhaps this could be discussed more fully.

It also appears that compared to 96h 27C, higher levels of HMR protein are detected after a shorter exposures of 27C (4-12h; Fig 2c), which may mean 12h is not sufficient time to reach the new phyBPfr:Pr steady state level, required to elicit a change in HMR levels, or that HMR protein stability is actually quite high and therefore not particularly sensitive to the diurnal changes in phyB action. In the latter scenario, HMR levels may be unaffected by phyB thermal regulation caused by increases in daytime temperatures (as suggested). So Figa/c may indicate the HMR-RCB thermal pathway may be essentially independent of phyB-control.

Co-IP assays showed that that HMR and RCB interact, and in vitro GST pulldowns mapped the interaction to HMR and RCB C-termini (Fig. 6). This is nice work, but since HMR appears to be more abundant in (persistent) cooler temperatures, it would be interesting to establish whether RCB levels rise in the warm and/or HMR and RCB only interact at when temperature increases. Do the authors have any insight into this?

A comment for inclusion in the discussion. The proposal shows HMR-RCB stimulates an increase in PIF4 abundance in response to increased temperature. However, PRRs have been shown to repress daytime PIF action, and POC1/PRR5 have been directly implicated in the repression of PIF4, restricting the response to specific times of day (Zhu et al., Nature Communications 2016 7, 13692).

Reviewer #2 (Remarks to the Author):

The manuscript entitled "RCB initiates Arabidopsis thermomorphogenesis by stabilizing the thermoregulator PIF4 in the daytime" by Qiu et al. describes a suppressor allele of the hmr-22 mutant rescuing all defects of the hmr-22 mutant in photomorphogenesis, chloroplast biogenesis and thermomorphogenesis including the stabilization of PIF4 in response to elevated temperature. Interestingly, the suppressor is a mutant allele of RCB1, a protein from the same class of nuclear/plastidic dual targeted factors like HMR.

The presented experimental data are sound, the text is well written and easy to follow. The content and design of the study follows the previous work by Qiu et al., 2019, Yoo et al., 2019 and Yang et al., 2019 and complements those papers that were all published in Nature Communications. The work is original, however, the study remains mainly descriptive, limiting its potential significance to the field of molecular plant biology.

The title, in my opinion, does not sufficiently reflect the content of the manuscript. It is not surprising, that RCB has comparable activity to HMR not only in photomorphogenesis and chloroplast development, as shown before, but also in stabilizing PIF4 during thermomorphogenesis. For that conclusion, it would have been sufficient to show that the rcb knock out mutant is deficient in thermomorphogenesis and PIF4 stabilization at high temperature. What the present manuscript rather suggests is that there might be a non-trivial functional interaction between HMR and RCB in the regulation of photo-and thermomorphogenesis. This

seems to me to be the most interesting aspect of the study, but at the same time it is also its biggest weakness. While the authors extensively and carefully describe the phenotypes of the *hmr-22* suppressor mutant (*rcb-101*), the study remains at the surface and there is no attempt to investigate the functional interaction of the two proteins. The most pressing question that needs to be addressed is, why only the RCB-A275V mutant can complement the HMR-D516N mutant phenotype, whereas the presence of the RCB wildtype protein has no effect, especially given the fact that the single knock out mutant versions of both proteins have very similar defects.

In the introduction the authors comprehensively discuss the importance of phytochrome B and the stabilization of PIF4 in light in the regulation of elongation growth in thermomorphogenesis during daytime. How can the findings about the HMR-RCB interaction be interpreted in this regard? As the authors mention, that the point mutations in HMR and RCB do not affect protein interaction with each other, do these maybe affect interaction with phyB or PIF4? What could be the mechanism linking HMR, RCB and phyB activity in regulating PIF4 abundance in light?

Please find some additional specific comments below:

1. page 11/ fig. 4d: "..., and PEP assembly was also rescued (Fig. 4d)"

Here it should be explained somewhere that *rpoB* (what is it?) and HMR were used as indicators for assembly of the PEP complex.

2. fig. 6g: why was the full length RCB not included in the pull-down assay?

3. page 14: "We did attempt to test whether these two mutations could alter the interaction between HMR and RCB. Neither the single mutations nor the combination of both led to observable changes in the HMR-RCB interaction in our experiments."

I think that data are important for the mechanistic interpretation and should be shown in the manuscript.

4. The HMR-D516N/RCB-A275V mutant pair functionally resembles the wildtype HMR-RCB pair, but the combination of HMR-D516N with RCB wildtype is not functional. Whereas the RCB-A275V mutant alone has wildtype phenotype, in the *hmr-22* background the *rcb-101* allele can restore PIF3 degradation, PIF4 accumulation and all related phenotypes. Why is it specific to the *rcb-101* allele and why is wildtype RCB not able to accomplish that? Can the authors provide a model/hypothesis about this and design experiments to test it?

Response to Reviewers

Reviewer #1

This latest paper by the Meng lab extends our understanding of an important mechanism that controls thermomorphogenesis in the model species Arabidopsis. The study complements a recent publication from the Meng lab which reported the discovery of REGULATOR OF CHLOROPLAST BIOGENESIS (RCB) as a nuclear/plastidial phytochrome signaling component required for plastid-encoded RNA polymerase (PEP) complex assembly (Yoo et al., Nature Communications 2019 10: 2629). This new submission describes the identification of a rcb-101 allele as a dominant suppressor of hmr-22 in thermo-regulation and proceeds to demonstrate that RCB operates with HEMERA (HMR), to control PIF3 signaling, and PIF4-mediated thermomorphogenesis. I feel that the data presented is of an excellent quality and largely supports the hypotheses proposed. I would like to offer the following comments:

The introductory and follow-on narrative could be tightened up a little. The authors firmly bring the reader's attention to the fact that phyB acts as a thermosensor (also final figure diagram), but it is difficult to link the potential loss of phyB(Pfr) activity at higher temperatures, with HMR-RCB module activity in thermomorphogenesis.

Response: We thank the reviewer for this comment. We have shortened the Introduction. We added the following sentence at the end of the second paragraph to emphasize the importance of understanding temperature signaling in the daytime when a significant amount of steady-state PHYB remains in the active form: “Because warm temperatures often coincide with high light intensities during the daytime – a combined light and temperature condition where a significant amount of steady-state PHYB remains in the active form¹² – the essence of understanding thermomorphogenesis is to elucidate how warm temperatures engage with PHYB signaling.”

The general perception of warm-temperature signaling remains that thermomorphogenesis is triggered by the loss of phyB activity (Pfr) -- which implies a similar mechanism to that of the shade responses. Although this mechanism is quite true for warm-temperature sensing at night, accumulating evidence suggests that temperature signaling during the daytime operates quite differently. For example, we recently reported that phyB-GFP exhibits distinct subnuclear localization patterns between warm-temperature and shade conditions, and it could still localize to large subnuclear photobodies that represent the active Pfr form [Hahm et al. (2020) *Nat Commun* 11:1660]. The results from this study provide evidence supporting an emerging theme that distinct mechanisms exist to selectively regulate individual PIFs in different combinations of light and temperature conditions. We have just begun to elucidate the underlying mechanism to stabilize PIF4 under warm temperatures in the light, which requires HMR and the newly uncovered temperature signaling component RCB.

The authors give the impression that PIF4 expression peaks at different times of day in SD and LD (an example from p6 is provided below), but in both LD and SD, PIF4 expression peaks at a similar time - around ZT8 (see Mockler datasets, <http://diurnal.mocklerlab.org/>).*

Response: The shift in the *PIF4* transcript level between LD and SD conditions was reported by Park et al. [*New Phytol* (2017) 215:269-280]. Also, from the website of the Mockler lab, if you plot the results of longday, SD in Ler, and SD in Col, you could clearly see the shift (shown below). The reviewer might refer to the “shortday” data, which somehow do not look as clear as the two SD results shown below.

It is (as the authors state) well established that in SDs, PIF4 transcript levels rise to higher levels during the long nights, than in LD. Further, because of this and as phyB activation promotes PIF4 degradation post-dawn, PIF4 protein is more abundant prior to dawn in SDs. This should be made clearer in the text. Also evident is that in SDs (as for LDs), PIF4 protein levels remain abundant during the light period (Yamashino et al., Plant Signal Behav 2013 Mar;8(3):e23390). The manuscript gives the impression that this is not the case.

Response: We thank the reviewer for the comment. We have revised the 4th paragraph in the Introduction to further clarify the timing of PIF4 accumulation and added the suggested reference.

Page 5 states “PHYB controls seedling morphogenesis primarily by regulating the stability of a family of basic helix-loop-helix transcription factors called PHYTOCHROME-INTERACTING FACTORS (PIFs)”. phyB is also known to regulate PIF activity through a sequestering mechanism – relevant to this study, this regulatory mechanism is proposed to be particularly important during the photoperiod (Park et al., Plant Cell. 2018 Jun;30(6):1277-1292).

Response: We totally agree that PHYB regulates the activity of PIFs besides their degradation. We modified the sentence to: “PHYB controls the activities of PIFs at multiple levels. During de-etiolation, photoactivated PHYB in the nucleus induces photomorphogenesis primarily by promoting ubiquitin-proteasome-dependent degradation of PIF1, PIF3, PIF4, and PIF5.”

Page 6 states “Notably, in the SD scenario, PIF4 expression is induced when PHYB has reverted to the inactive Pr. Therefore, the regulation of PIF4 stability by PHYB is not a major mechanism in thermomorphogenesis under SD conditions.” Could this be explained a bit more fully.

Response: We have revised this paragraph, but the concept stays the same. In SD conditions, *PIF4* transcripts accumulate at the end of the night, when most PHYB has converted to the inactive Pr. This is supported by the fact that photobodies -- an indicator of the Pfr form of PHYB -- become almost invisible between 12-18h during a light-to-dark transition (Van Buskirk et al. 2014, *Plant Physiol* 165:595-607). So, when PIF4 accumulates before dawn in SD conditions, it is mainly due to the upregulation of PIF4 transcripts as opposed to the regulation of active-PHYB-mediated PIF4 degradation.

**On page 6 the authors state: “because PIF4 transcripts peak during the daytime in LD, PIF4 transcription is negatively regulated by the transcriptional regulator ELONGATED HYPOCOTYL 5 (HY5)⁴⁵”. This could give the impression that HY5 just operates in LDs, however, to my knowledge the hy5 mutant is effective in both SD, 12:12 and LD, implying HY5 operates across photoperiods (e.g. Ang and Deng, *Plant Cell* 1994 6, 613-628; Toledo-Ortiz *Plos Genet* 2014, 10;6, e1004416; Gangappa and Kumar *Cell Rep.* 2017 18(2):344-351).*

Response: We have removed this part to make the Introduction more concise. What we referred to in the previous version was HY5’s role in the regulation of PIF4 in the context of temperature signaling. We agree that HY5 plays a major role in light signaling in both LD and SD conditions. But, HY5’s role in the regulation of the *PIF4* transcript level is more relevant in LD based on the following two references: Delker et al. 2014, *Cell Rep* 9:1983-9 and Gangappa et al. 2017, *Cell Rep* 18:344-351.

Page 6 states “under LD, in striking contrast to SD conditions, because PIF4 needs to accumulate in the light when PHYB is in the active Pfr, a critical mechanism must be implemented to stabilize PIF4 or antagonize PHYB-mediated PIF4 degradation”. Maybe a slight rewording required since the authors have already suggested that phyB is proposed to be less active in warm conditions, (so maybe there is less of a requirement to “antagonize” phyB action).

Response: The view that PIF4 is prominently degraded by active PHYB in warm temperatures is supported by the following evidence. First, PHYB-FP assembles to large photobodies -- an indicator of the Pfr form of PHYB -- even in elevated temperatures, suggesting that a significant amount of PHYB remains in the Pfr state [Hahm et al. (2020) *Nat Commun* 11:1660]. Second, PIF4 accumulates to a much higher level in *phyB-9* compared to Col-0 in warm temperatures,

indicating that PIF4 is actively degraded by PHYB in elevated temperatures [Fiorucci et al. (2020) *New Phytol* 226:50-58]. We agree with the reviewer and have changed “antagonize” to “modulate”. We revised the sentence to “Because, even in elevated temperatures, a significant amount of PHYB during the daytime stays in the Pfr form that mediates PIF4 degradation^{12,45}, a mechanism must be implemented to stabilize PIF4 or modulate PHYB-mediated PIF4 degradation.”

Comments on the data.

Fig 2a shows HMR protein levels are reduced after 96h at 27°C compared to 21°C. As the authors suggest, this is consistent with expected increase in phyB thermal reversion at the warmer 27°C temperature. The lab’s published data showed phyBPfr promotes HMR protein accumulation (Rafaelo 2012), so increased thermal reversion may reduce steady state Pfr levels, and lead to a reduction in HMR. Despite the lower HMR levels in 27°C compared to 21°C, HMR (and RCB) appears to be important for controlling PIF4 protein levels at 27°C and not 21°C (Fig 2a). This data is slightly counterintuitive and may indicative of another temperature-regulated mechanism at play. Another thing to consider is whether the temperature shift to 27°C is sufficient to reduce phyBPfr:Pr steady state levels at the red fluence rates used (50 umol). Perhaps this could be discussed more fully.

It also appears that compared to 96h 27C, higher levels of HMR protein are detected after a shorter exposures of 27C (4-12h; Fig 2c), which may mean 12h is not sufficient time to reach the new phyBPfr:Pr steady state level, required to elicit a change in HMR levels, or that HMR protein stability is actually quite high and therefore not particularly sensitive to the diurnal changes in phyB action. In the latter scenario, HMR levels may be unaffected by phyB thermal regulation caused by increases in daytime temperatures (as suggested). So Figa/c may indicate the HMR-RCB thermal pathway may be essentially independent of phyB-control.

Response: We thank the reviewer for the comment. We have revised the paragraph to emphasize the point that the rescue of PIF4 accumulation in *rcb-101/hmr-22* is not due to changes in the level of HMR. The reduced level of HMR is intriguing. It could be due to the overall reduction of PHYB activity. The photoreversion rate should stay constant at a given fluence rate of light, so an increase in the thermal reversion rate by raising the temperature theoretically should destabilize the Pfr. Alternatively, the decrease in HMR could due to a warm-temperature response in chloroplasts. We still do not know much about the temperature effects on chloroplasts or chloroplast transcription. We are measuring the overall HMR level, including chloroplast-localized HMR that is the same size as the nuclear HMR [Nevarez et al. (2017) *Plant Physiol* 173:1953-66]. Therefore, it is possible that the change in the total level of HMR in warm temperatures is also contributed by changes in the level of chloroplast HMR. We did not follow up on this here because there is no difference in the level of HMR among the lines examined, suggesting that altering the level of HMR is unlikely the cause for the phenotypes of *hmr-22* and *rcb-101/hmr-22*.

Co-IP assays showed that that HMR and RCB interact, and in vitro GST pulldowns mapped the interaction to HMR and RCB C-termini (Fig. 6). This is nice work, but since HMR appears to be more abundant in (persistent) cooler temperatures, it would be interesting to establish whether RCB levels rise in the warm and/or HMR and RCB only interact at when temperature increases. Do the authors have any insight into this?

Response: Our goal was to first demonstrate that RCB interacts with HMR in vivo and in vitro, which is strongly supported by the genetic data. We had hoped that we could detect any changes in the RCB-HMR interaction caused by the mutations in *hmr-22* or *rcb-101* to explain the cause of the phenotypes of *rcb-101/hmr-22*, but we did not observe any obvious changes. In our preliminary studies, we did not observe a significant change in the RCB level by warm temperatures. Based on the results that HMR and RCB are required for photomorphogenesis at 21 °C and thermomorphogenesis at 27 °C, we reasoned that we should be able to detect the HMR-RCB interaction in both conditions. We did not attempt to measure temperature-dependent changes in the HMR-RCB interaction using Co-IP, because the Co-IP experiments require grinding the tissue in liquid nitrogen and subsequent incubation of the protein extract with antibody-conjugated beads at 4 °C for an extended period of time, any temperature-dependent changes in HMR-RCB interactions in vivo would most likely be disrupted during the experimental process. For our experiments, we used seedlings grown at 21 °C for the Co-IP experiments to show that RCB and HMR are associated with each other in vivo. The rationale behind this experiment was that RCB and HMR are both required for light signaling in 21 °C.

A comment for inclusion in the discussion. The proposal shows HMR-RCB stimulates an increase in PIF4 abundance in response to increased temperature. However, PRRs have been shown to repress daytime PIF action, and POC1/PRR5 have been directly implicated in the repression of PIF4, restricting the response to specific times of day (Zhu et al., Nature Communications 2016 7, 13692).

Response: We have added the role of TOC1/PRR5 in the regulation of PIF4 activity and the reference in the Discussion.

Reviewer #2:

*The manuscript entitled "RCB initiates Arabidopsis thermomorphogenesis by stabilizing the thermoregulator PIF4 in the daytime" by Qiu et al. describes a suppressor allele of the *hmr-22* mutant rescuing all defects of the *hmr-22* mutant in photomorphogenesis, chloroplast biogenesis and thermomorphogenesis including the stabilization of PIF4 in response to elevated temperature. Interestingly, the suppressor is a mutant allele of RCB1, a protein from the same class of nuclear/plastidic dual targeted factors like HMR.*

The presented experimental data are sound, the text is well written and easy to follow. The content and design of the study follows the previous work by Qiu et al., 2019, Yoo et al., 2019 and Yang et al., 2019 and complements those papers that were all published in Nature Communications. The work is original, however, the study remains mainly descriptive, limiting its potential significance to the field of molecular plant biology.

The title, in my opinion, does not sufficiently reflect the content of the manuscript. It is not surprising, that RCB has comparable activity to HMR not only in photomorphogenesis and chloroplast development, as shown before, but also in stabilizing PIF4 during thermomorphogenesis. For that conclusion, it would have been sufficient to show that the rcb knock out mutant is deficient in thermomorphogenesis and PIF4 stabilization at high temperature.

Response: We thank the reviewer for the comments. The identification of a missense *rcb* allele that rescues a weak allele of *hmr* was completely unexpected to us. This is not only because the *hmr-22* suppressor screen was performed in parallel with the tall-and-albino seedling screen that identified the null *rcb* alleles [Yoo et al. (2019) *Nat Commun* 10:2629], but also because, even with the knowledge of RCB in PHYB signaling, we could not have foreseen by any stretch of the imagination that a missense mutation in RCB could have an opposite effect and rescue the defects of the *hmr-22* allele in an allele-specific manner -- which uncovers an intimate functional relationship between RCB and HMR in PHYB signaling.

Although *rcb-101* rescues all defects of *hmr-22*, we chose to use the title to highlight the discovery of RCB as a novel signaling component in thermomorphogenesis and its essential role in PIF4 stabilization, because: (1) the original screen was aimed at identifying new signaling component in daytime thermomorphogenesis, whose mechanism remains poorly understood, and (2) we wanted to highlight the concept of selective regulation of PIF4 by warm temperatures during the daytime. When we first reported the role of HMR in thermomorphogenesis, it was quite surprising (to us and many colleagues) that HMR plays opposing roles in the regulation of PIF3 and PIF4 -- HMR facilitates PIF3 degradation but promotes PIF4 stabilization. This study further demonstrates the two distinct effects on PIF3 and PIF4 and identifies RCB as a new component in this signaling mechanism. Although we still do not understand the detailed biochemical mechanism by which HMR and RCB regulate PIF3 and PIF4, this identification of RCB as a novel temperature signaling component and its collaborative role with HMR opens a completely new avenue and illuminates a clear path for investigating the underlying mechanism.

*What the present manuscript rather suggests is that there might be a non-trivial functional interaction between HMR and RCB in the regulation of photo- and thermomorphogenesis. This seems to me to be the most interesting aspect of the study, but at the same time it is also its biggest weakness. While the authors extensively and carefully describe the phenotypes of the *hmr-22* suppressor mutant (*rcb-101*), the study remains at the surface and there is no attempt to investigate the functional interaction of the two proteins. The most pressing question that needs to be addressed is, why only the RCB-A275V mutant can complement the HMR-D516N mutant phenotype, whereas the presence of the RCB wildtype protein has no effect, especially given the fact that the single knock out mutant versions of both proteins have very similar defects.*

In the introduction the authors comprehensively discuss the importance of phytochrome B and the stabilization of PIF4 in light in the regulation of elongation growth in thermomorphogenesis during daytime. How can the findings about the HMR-RCB interaction be interpreted in this regard? As the authors mention, that the point mutations in HMR and RCB do not affect protein interaction with each other, do these maybe affect interaction with phyB or PIF4? What could be the mechanism linking HMR, RCB and phyB activity in regulating PIF4 abundance in light?

Response: We agree with the reviewer that this study unveils an important signaling mechanism -- the HMR-RCB interaction -- which enables PIF4 accumulation for thermomorphogenetic responses. The genetic and biochemical evidence supports the intimate functional relationship between RCB and HMR. But, our current data still do not offer an explanation for the specific rescue of *hmr-22* by *rcb-101*. We agree that there are so many possibilities, as suggested by the reviewer, because the mechanism of PIF4 degradation/stabilization and the function of HMR in PHYB signaling and subnuclear localization are still not fully understood. For example, although it has been shown that PHYB controls the stability of PIF3 and PIF4, we have only recently clarified that the regulation of PIF3 degradation is through the C-terminal module of PHYB as opposed to the N-terminal photosensory module suggested previously [Qiu et al. (2017) *Nat Commun* 8:1905]. How the C-terminal module of PHYB interacts with PIFs and regulates their stability remains poorly understood. With the previously unknown RCB-HMR link, the results of this study will enable further investigations on the regulation of the stability of PIF3 and PIF4 by PHYB, HMR, and RCB in diverse light and temperature conditions.

Please find some additional specific comments below:

1. page 11/ fig. 4d: "..., and PEP assembly was also rescued (Fig. 4d)"

Here it should be explained somewhere that rpoB (what is it?) and HMR were used as indicators for assembly of the PEP complex.

Response: We thank the reviewer for the suggestion. We added the explanation of using *rpoB* and HMR as indicators for the assembly of the PEP complex: "Consistently, the 1000-kD PEP complex in *hmr-22* was not detectable by antibodies against either a core PEP component, RPOB, or a PEP accessory protein, HMR (Fig. 4d)."

2. fig. 6g: why was the full length RCB not included in the pull-down assay?

Response: We have previously shown that both the nuclear and chloroplast RCB proteins have the same size as the mature form (without the N-terminal transit peptide) [Yoo et al. (2019) *Nat Commun* 10:2629]. The full-length RCB pull-down data are shown in Fig. 6f. In Fig. 6g, we aimed to identify domains involved in the HMR interaction in the mature RCB. We added the explanation in the text: "To examine which region of RCB is involved in the HMR interaction, we used GST-HMR Δ 251 as the bait to pull down three *in vitro* translated, HA-tagged truncation

fragments of RCB (Fig. 6e): HA-RCB Δ 51, which lacks the transit peptide, mimicking the endogenous RCB²⁷”.

3. page 14: “We did attempt to test whether these two mutations could alter the interaction between HMR and RCB. Neither the single mutations nor the combination of both led to observable changes in the HMR-RCB interaction in our experiments.”

I think that data are important for the mechanistic interpretation and should be shown in the manuscript.

Response: We added a new Supplementary Fig. 2 to show that we could not detect observable changes in the RCB-HMR interaction by the D516N mutation in HMR22.

4. *The HMR-D516N/RCB-A275V mutant pair functionally resembles the wildtype HMR-RCB pair, but the combination of HMR-D516N with RCB wildtype is not functional. Whereas the RCB-A275V mutant alone has wildtype phenotype, in the hmr-22 background the rcb-101 allele can restore PIF3 degradation, PIF4 accumulation and all related phenotypes. Why is it specific to the rcb-101 allele and why is wildtype RCB not able to accomplish that? Can the authors provide a model/hypothesis about this and design experiments to test it?*

Response: These are great questions. We agree that we should focus on solving this puzzle next. As we stated in the Discussion, there are two major possibilities. First, it is still possible that the mutations in HMR22 and RCB101 affect the HMR-RCB interaction directly but mildly and beyond detection by our current experimental approaches. Alternatively, the mutations may not affect the RCB-HMR interaction *per se* but rather alter the activity of the HMR-RCB complex. For example, while an RCB-HMR22 complex is dysfunctional, the RCB101-HMR and RCB101-HMR22 could be as functional as the wildtype RCB-HMR complex. In the latter model, the mutations might impact the interaction between the HMR-RCB complex and another signaling molecule such as PHYB or PIF3/4. As indicated in the response to Question #1, testing these hypotheses would require a deeper understanding of the PHYB-PIF interaction involved in the regulation of the stability of PIF3 and PIF4 first, and then, the roles of RCB and HMR.

REVIEWERS' COMMENTS

Reviewer #2 (Remarks to the Author):

The revised version of the manuscript does not include new data (except for one SI figure). The authors mainly discussed the points raised by the reviewers in the rebuttal letter and performed only minor text changes.

As stated in the first review, without additional data providing insight into the functional interaction of HMR and RCB, or the mechanism of RCB action in PIF4 stabilization during thermomorphogenesis, I still consider the study of limited significance to the field.

Additional points:

I noticed that the authors have changed the name of the suppressor mutant from hms to rcb-101 (line 133) but find this a bit confusing because at this stage it was actually not known that the suppressor was a rcb-allele.

Further I agree with reviewer 1 who pointed out it is questionable whether temperature rise to 27 °C affects phyB Pfr levels at a high fluence rate of 50 $\mu\text{mol m}^{-2} \text{s}^{-1}$. Data from Legris et al. 2016 or Hahm et al. 2020 rather suggest no Pfr reduction at high fluence rates, demonstrating that the photoconversion rate is still faster compared with the thermal reversion rate at 27 °C. The authors did not discuss this adequately.

In line 281-283 the authors say that: "We did attempt to test whether these two mutations could alter the interaction between HMR and RCB. However, our experiments did not detect any observable changes in the HMR-RCB interaction caused by the D516N mutation in hmr-22 (Supplementary Fig. 2)." But then only showed data for the D516N mutation in HMR but not for the A275V mutation in RCB.

Response to Reviewer Comments

Reviewer #2

Additional points:

I noticed that the authors have changed the name of the suppressor mutant from hms to rcb-101 (line 133) but find this a bit confusing because at this stage it was actually not known that the suppressor was a rcb-allele.

Response: We thought that keeping the suppressor mutant the same name throughout the article makes it easier for readers to follow. We have edited the text at the beginning of the Results by indicating that the name of the suppressor mutant will be further explained later in the text.

Further I agree with reviewer 1 who pointed out it is questionable whether temperature rise to 27 °C affects phyB Pfr levels at a high fluence rate of 50 $\mu\text{mol m}^{-2} \text{s}^{-1}$. Data from Legris et al. 2016 or Hahm et al. 2020 rather suggest no Pfr reduction at high fluence rates, demonstrating that the photoconversion rate is still faster compared with the thermal reversion rate at 27 °C. The authors did not discuss this adequately.

Response: We thank the reviewer for the comment. We have added a sentence in the Results to explain the rationale behind our experimental design, and added a new paragraph in the Discussion to discuss the implications of the results in the relatively high light condition. The rate of thermal reversion is a biophysical constant of the PHYB molecule and is not influenced by light intensity -- i.e., in either low or high intensity of R light (under the same temperature), the thermal reversion rate of PHYB remains constant. So, the acceleration of thermal reversion by warm temperature should have the same impact on the Pfr/Pr equilibrium regardless of light intensity. PHYB-mediated PIF4 degradation should operate in both low and high light conditions -- in *phyB-9*, PIF4 accumulates to much higher levels than Col-0 in warm temperatures even under strong white light conditions (Fiorucci et al. 2020 *New Phytol* 226:50-58). Therefore, although it has been suggested, based on modeling, that the effect of warm temperature on PHYB should best observed in low light conditions, our results here, combined with the previously published data (Qiu et al. 2019 *Nat Commun*), show that PHYB-mediated PIF4 degradation is significantly reduced even under high intensities of red light, providing experimental evidence that, although the rate of thermal reversion of PHYB might not be theoretically higher than that of photoreversion, warm temperature can significantly attenuate PHYB signaling even in high light intensities.

In line 281-283 the authors say that: "We did attempt to test whether these two mutations could alter the interaction between HMR and RCB. However, our experiments did not detect any observable changes in the HMR-RCB interaction caused by the D516N mutation in hmr-22 (Supplementary Fig. 2)." But then only showed data for the D516N mutation in HMR but not for the A275V mutation in RCB.

Response: We have revised the sentence to “An obvious hypothesis would be that RCB101 rescues a defect in the interaction between HMR22 and the wild-type RCB. However, our experiments did not detect any observable changes in the HMR-RCB interaction caused by the D516N mutation in HMR22 (Supplementary Fig. 2).”